# Global patterns of plant functional traits and their relationships to climate
Jiaze Li [1] & Iain Colin Prentice [1,2]

Plant functional traits (FTs) determine growth, reproduction and survival strategies of plants adapted to their growth environment. Exploring global geographic patterns of FTs, their covariation and their relationships to climate are necessary steps towards better-founded predictions of how global environmental change will affect ecosystem composition. We compile an extensive global dataset for 16 FTs and characterise trait-trait and trait-climate relationships separately within non-woody, woody deciduous and woody evergreen plant groups, using multivariate analysis and generalised additive models (GAMs). Among the six major FTs considered, two dominant trait dimensions—representing plant size and the leaf economics spectrum (LES) respectively—are identified within all three groups. Size traits (plant height, diaspore mass) however are generally higher in warmer climates, while LES traits (leaf mass and nitrogen per area) are higher in drier climates. Larger leaves are associated principally with warmer winters in woody evergreens, but with wetter climates in non-woody plants. GAM-simulated global patterns for all 16 FTs explain up to three-quarters of global trait variation. Global maps obtained by upscaling GAMs are broadly in agreement with iNaturalist citizen-science FT data. This analysis contributes to the foundations for global trait-based ecosystem modelling by demonstrating universal relationships between FTs and climate.

As primary producers, land plants regulate carbon and nitrogen cycling in ecosystems and determine their productivity[1-3]. Functional traits (FTs)—morphological, physiological, and phenological characteristics of individual plants[4]—determine plant growth, reproduction, and survival strategies by influencing life-history processes (e.g., seed traits), resource uptake and utilisation (e.g., leaf nutrient contents), and responses to environmental change and disturbances (e.g., leaf traits and plant height)[4-6]. FTs thus reflect the diverse ecological strategies of plants, which influence plant community assembly and thereby biodiversity and function[5,7-9].

FTs have been found to covary along a limited number of dimensions[4,10]. FTs that vary together at species level might, in principle, share genetic controls, or depend on trade-offs representing alternative ecological strategies[10]. The leaf economics spectrum (LES) is one well-known axis of trait covariation that embodies a trade-off between resource acquisition and conservation[11]. Díaz and colleagues[12] summarised the global spectrum of plant form and function based on six major FTs that are critical for the growth, survival, and reproduction of species. They identified two orthogonal dimensions of trait covariation: one corresponding to the size of whole plants and plant organs, the other to the LES[12]. These two trait dimensions are universal, existing even in ecosystems with extreme climates,

such as tundra[13]; at both species and community scales[9,14]; and within plant groups, e.g., trees[15]. Different ecological strategies can facilitate the coexistence of species (and their FTs) within communities[9]. At the community level, covarying FTs tend to perform interrelated functions[2,16-18]. Wright et al.[16] observed that wood density and leaf size are negatively correlated in neotropical forests. Lower wood density related to higher water conductivity per unit sapwood area and larger leaf area associated with potentially higher photosynthetic rate but also higher evapotranspiration. These two traits are thus interconnected through the hydraulic system governing water transport and its trade-off with photosynthetic efficiency[16]. Quantifying the covariation of FTs and their controls is necessary for a full understanding of how the diversity of plants translates into community composition, productivity, and adaptations to environment[10].

Although the two dimensions of trait variation identified in ref. 12 are universal, the prevalence of specific FTs varies along environmental gradients[2,19]: different combinations of FTs are selected for by environmental conditions[5,14,20], so climate change is expected to bring about shifts in species composition determined by their FTs, such as migration and extinction of needle-leaved trees in part of North America since the last

[1]Georgina Mace Centre for the Living Planet, Department of Life Sciences, Imperial College London, Silwood Park Campus, Buckhurst Road, Ascot, SL5 7PY, UK. [2]Department of Earth System Science, Ministry of Education Key Laboratory for Earth System Modeling, Institute for Global Change Studies, Tsinghua University, Beijing, 100084, China. ✉e-mail: jiaze.li19@imperial.ac.uk

**Table 1 | Plant functional traits used in this study**

| Trait[a] | Abbreviation[b] | Unit[c] | Functions[d] | Availability[e] |
|---|---|---|---|---|
| Leaf area | LA | mm$^2$ | Leaf size; related to leaf energy and water balance[10] | Plot-level trait means[38] and species-level trait means[40] |
| Leaf mass per unit area | LMA | kg/m$^2$ | Key traits in the leaf economic spectrum (LES); reflecting investment strategy[11] | |
| Leaf nitrogen content per unit area | $N_{area}$ | g/m$^2$ | | |
| Stem specific density | SSD | g/cm$^3$ | Trait in wood economics spectrum; reflecting drought-tolerance[70] | |
| Plant height | H | m | Whole plant size; reflecting the ability to compete for light[12,54] | |
| Diaspore mass | DM | mg | Seed size; reflecting dispersal and establishment ability[12,55] | |
| Leaf fresh mass | LFM | g | Related to energy transfer, plant growth and productivity[9] | Plot-level trait means[38] |
| Leaf phosphorus content per unit area | $P_{area}$ | g/m$^2$ | | |
| Leaf carbon content | $C_{mass}$ | mg/g | | |
| Leaf dry matter content | LDMC | g/g | | |
| Stem conduit (vessel and tracheid) element length | WVL | µm | Related to water transport: safety versus efficiency[9] | |
| Stem conduit density | SCD | mm$^{-2}$ | | |
| Seed number per reproductive unit | SN | NA | Reflecting dispersal, regeneration and reproduction ability[9]. Here, SL is the length of a seed, while DUL is the length of the dispersal unit, which may include many seeds. | |
| Seed length | SL | mm | | |
| Dispersal unit length | DUL | mm | | |
| Leaf nitrogen isotope ratio | $\delta^{15}N$ | per million | Reflecting nitrogen source, loss pathways, and root symbionts[9,94] | |

[a]Plant functional traits used in this study. [b]Abbreviation and [c]unit of each of the 16 traits. [d]Brief description of functions for each of the 16 traits. [e]Data availability for each trait. All 16 plant functional traits have plot-level trait means (community-weighted means, CWMs)[38], while the first six traits also have species-level mean trait values[40].

glacial maximum[21]. In order to predict such shifts, it is essential to understand global trait-climate relationships[2,21]. But there is no comprehensive global map of trait measurements[22]. The models used most widely to investigate climate change effects on ecosystems are Dynamic Global Vegetation Models (DGVMs). These typically compress functional diversity into a small (about 5–15) number of plant functional types (PFTs)[22,23], which are assigned fixed trait values. Substantial, observed variability within PFTs is thereby lost[24,25]. Moreover, as past and future climates may have no analogues in present climate conditions, it is likely that this approach misses potential trait variation and trait combinations within PFTs[24,26,27]. A better approach to estimating trait values on a global scale is needed to overcome the limitations of PFT-based modelling. Although satellite remote-sensing offers potential to directly map plant traits at large scales[28–30], this approach can only retrieve a limited set of traits from leaves and upper canopies, with moderate accuracy[28]. In addition, statistical upscaling and machine learning approaches based on relationships between FTs and environmental factors have recently been used to produce global maps for a few (mainly leaf) FTs[2,3,25,29,31–34]. A recently published work simulated global maps of three leaf traits via optimality models based on eco-evolutionary optimality theory[35]. However, the various published global trait maps do not always show consistent global patterns, reflecting differences in data sources and upscaling methods[35,36].

Newly published data and computational resources should facilitate global-scale research on FTs. The sPlotOpen dataset[37,38], a curated subset of a large globally distributed set of vegetation plots, provides plot-level trait data (as community-weighted means, CWMs) for 18 FTs. These CWMs were calculated from trait data in the TRY database[39] and the relative abundances of each taxon within each plot[38]. Díaz et al.[40] recently published an enhanced species-level trait dataset containing the species-mean values of the six major FTs defining the primary axes of trait covariation in the global spectrum of plant form and function as defined in ref. 12 Global trait patterns have also been mapped by complementing plant observations from the global citizen-science project iNaturalist with measurements from TRY[41].

A study of the geographical patterns of six FTs across North and South America found that trait-climate relationships differ overall between woody and herbaceous plants, including different climate predictors and different response shapes of FTs along climatic gradients[42]. Previous studies have also found that woody evergreen plants tend to have thicker and longer-lived leaves poleward, while woody deciduous plants show the opposite pattern[43,44]. A recently developed theory explains this divergence of latitudinal trends[45]. According to the theory, leaves maximise the average net carbon gain over their life cycle. The different relationships between leaf mass per area (LMA) and environmental factors in deciduous and evergreen woody plants then arise because the life cycle of deciduous leaves, unlike that of evergreen leaves, is tied to the annual cycle[45]. It is, therefore, appropriate to analyze non-woody plants, woody deciduous plants, and woody evergreen plants as separate groups with potentially distinct trait-environment relationships.

We compiled a global dataset for 16 FTs (Table 1), including 42,676 plant taxa from 77,074 natural vegetation plots based on the sPlotOpen dataset[38] and the enhanced species-level trait dataset of ref. 40. All FTs in our dataset have plot-level means for each plot; six also have species-level means. We refer to these six as 'major FTs' as they also form the global spectrum of plant form and function. Plant taxa were grouped into non-woody, woody deciduous and woody evergreen categories according to their life form and leaf phenology. We analysed trait covariation and characterised trait-climate correlations within each category for the six major FTs. For all FTs, we fitted generalised additive models (GAMs) for their relationships to three bioclimatic variables. Then we simulated global patterns for each FT at 0.1° spatial resolution, based on climate data and remotely sensed global fractional cover of the three plant groups. Finally, we compared our global maps for the 16 FTs against the iNaturalist maps[41] at 2° spatial resolution.

## Results
### Trait combinations
We calculated CWMs of each of the three plant groups within each vegetation plot for the six major FTs. CWMs were natural-log transformed prior to further analysis. We then used the CWMs to evaluate covariation among

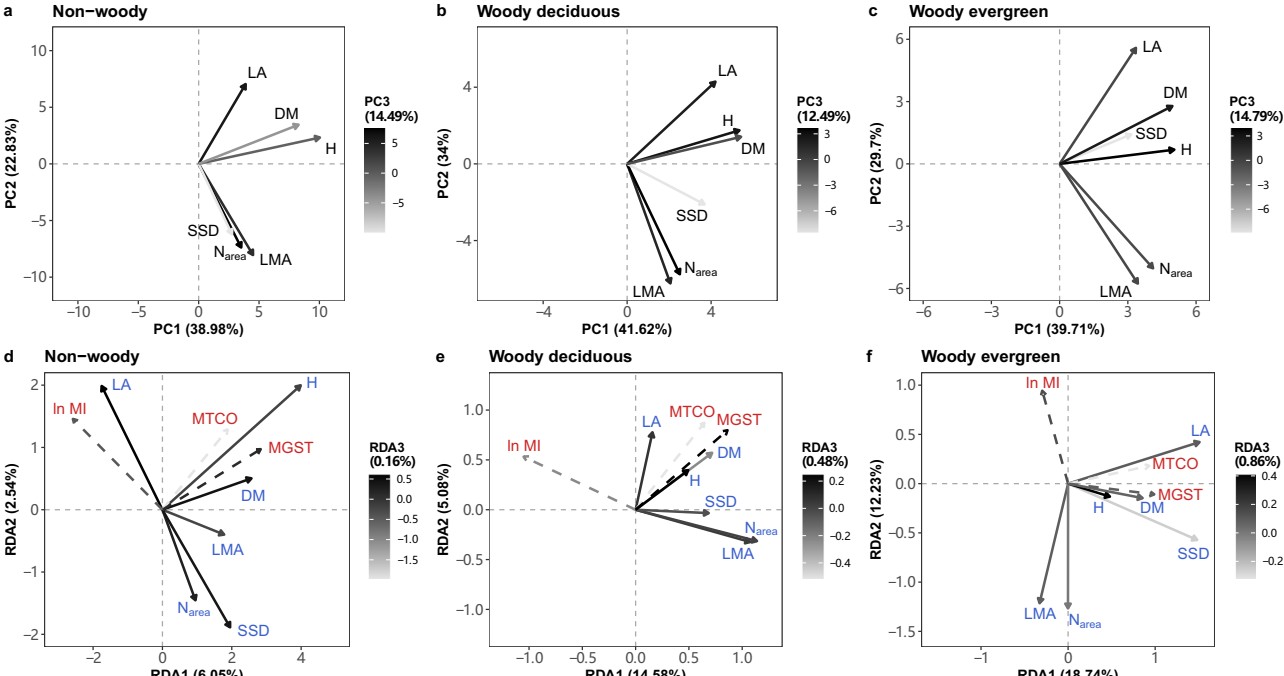

**Fig. 1 | Trait-trait and trait-climate correlations.** Principal component analysis (PCA) (**a**–**c**) and redundancy analysis (RDA) (**d**–**f**) for six major plant functional traits of non-woody (**a**, **d**), woody deciduous (**b**, **e**) and woody evergreen (**c**, **f**) plants. Six major traits are leaf area (LA, mm²), leaf mass per unit area (LMA, kg/m²), leaf nitrogen content per unit area ($N_{area}$, g/m²), stem specific density (SSD, g/cm³), plant height (H, m) and diaspore mass (DM, mg). The orientation of axes has been rotated according to the Fig. 2 in ref. 12. Grey scales indicate the loadings on the third axis. Solid arrows (with blue labels in RDA) represent traits, and dashed arrows (with red

labels in RDA) represent bioclimatic variables. All six traits were natural-log transformed before analysis. Both log-transformed traits and bioclimatic variables were rescaled to a mean of 0 and a standard deviation of 1 before analysis. ln MI, log-transformed moisture index; MTCO mean temperature of the coldest month, MGST mean growing-season temperature (see Methods for definition). All elements of the figure were created by the authors using R v4.2.2 (https://www.r-project.org/)[81].

FTs within the non-woody, woody deciduous and woody evergreen groups. Patterns of covariation were summarised by Principal Component Analysis (PCA). The first three PCA axes together captured 76% of trait variation in non-woody taxa, 88% in woody deciduous taxa and 84% in woody evergreen taxa (Fig. 1a–c, Table S6). Two orthogonal dimensions emerged for all groups. One dimension, represented by height (H) and diaspore mass (DM), represents overall plant size—from short plants with light diaspores to tall plants with heavy diaspores. The other, represented by LMA and leaf nitrogen per unit area ($N_{area}$), reflects the LES[11] and the predominant influence of LMA on $N_{area}$[46,47]. Leaf area (LA) tended to covary with plant size, while being approximately orthogonal to the LES. Stem-specific density (SSD) showed the opposite direction to the other traits along axis 3 of PCA. SSD in non-woody plants was strongly correlated with LMA and $N_{area}$ (Fig. 1a). SSD in woody evergreen plants was more correlated with plant size (Fig. 1c), while SSD in woody deciduous plants showed an intermediate pattern (Fig. 1b).

### Relationships between traits and climate

We calculated three bioclimatic variables for each plot (Fig. S2), which represent the three major independent climatic controls on the global geographic distribution of vegetation physiognomy[48,49,50] and therefore likely also the global distribution of key plant traits[48] (see Methods). The three bioclimatic variables are: mean growing-season temperature (MGST, °C, representing summer warmth), mean temperature of the coldest month (MTCO, °C, representing winter cold), and the log-transformed moisture index (ln MI, unitless, representing plant-available moisture: see Methods). We subsequently conducted a Redundancy Analysis (RDA) to describe the extent to which the six major FTs covary with these bioclimatic variables. The first three constrained RDA axes accounted for 9%, 20% and 32% of total trait variation in non-woody, woody deciduous

and woody evergreen plants, respectively (Fig. 1d–f, Table S7). These three axes revealed substantially the same combinations of plant size and LES traits as were shown in the PCA, but further illustrated their relationships with bioclimate. Along the three axes (Fig. 1d–f), plant height and diaspore mass increased with the growing-season temperature (MGST) within all groups, and LMA and $N_{area}$ decreased with increasing moisture (ln MI) within all groups. Plant size traits were positively correlated with winter cold (MTCO) on the first two axes, but they were negatively correlated on the third axis. However, LA behaved differently, covarying with ln MI in non-woody plants but with temperature variables in woody plants. The residual or unconstrained RDA axes (Fig. S6) approximately replicate the PCA axes, indicating that much trait variation along the main axes identified in ref. 12 persists even after climatic influences have been taken into account. However, a strong correlation between SSD and plant size traits in woody evergreen plants disappeared (Fig. S6c).

We used GAMs to quantify relationships between each major FT and bioclimatic variables in more detail, which allowed us to fit more complex surfaces and to distinguish growing season from cold-month temperature effects (Figs. 2–3). We calculated the relative importance values of explanatory variables (Fig. 2) in GAMs to quantify the individual contribution of three bioclimatic variables to the major FTs. These values measure the average percentage contribution of each variable in turn to the fit of the models based on all three variables. Across all plant groups H variation was dominated by growing-season temperature, and LMA and $N_{area}$ variation by moisture availability (with an additional influence of growing-season temperature on LMA in non-woody plants). DM variation was dominated by moisture availability in non-woody plants but by growing-season temperature (with an additional influence of winter temperature) in woody deciduous plants. By contrast, variation in SSD and LA showed similar controls in non-woody and woody deciduous plants (dominated by

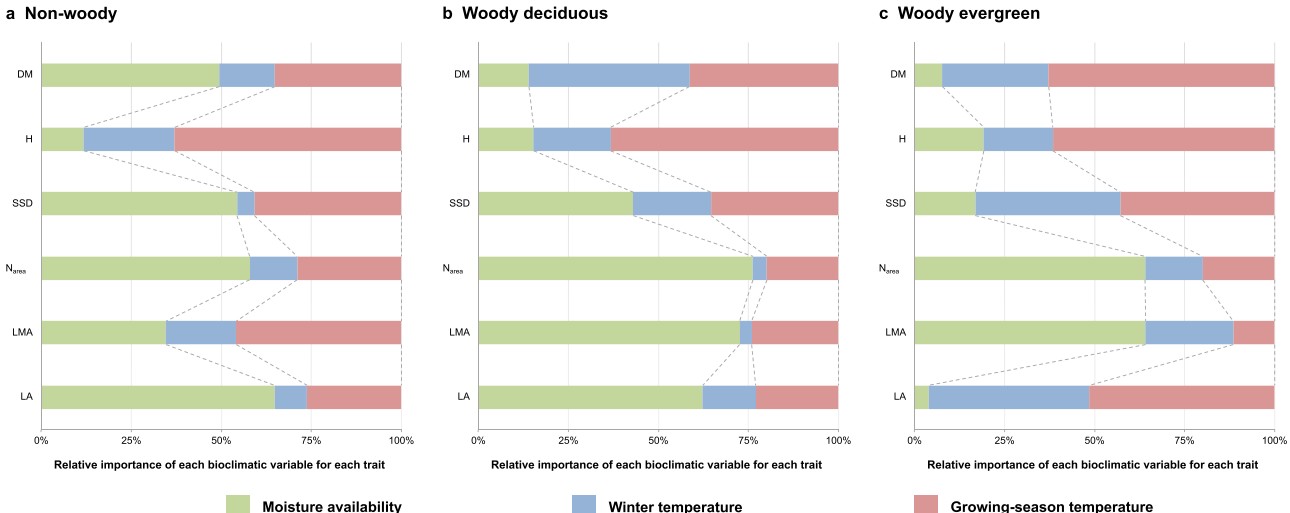

**Fig. 2 | Relative importance of three bioclimatic variables.** Relative importance of each bioclimatic variable in predicting six major plant functional traits of non-woody (**a**), woody deciduous (**b**) and woody evergreen (**c**) plants derived from Generalised Additive Models (GAMs). Six major traits are leaf area (LA, mm$^2$), leaf mass per unit area (LMA, kg/m$^2$), leaf nitrogen content per unit area ($N_{area}$, g/m$^2$), stem specific density (SSD, g/cm$^3$), plant height (H, m) and diaspore mass (DM, mg). All six traits were natural-log transformed before the analysis. Moisture availability, log-transformed moisture index (ln MI, unitless); winter temperature, mean temperature of the coldest month (MTCO, °C); growing-season temperature, mean growing-season temperature (MGST, °C) (see Methods for definition). All elements of the figure were created by the authors using R v4.2.2 (https://www.r-project.org/)[81].

moisture availability and growing-season temperature for SSD and moisture availability for LA), but a different type of control—dominated by growing-season and winter temperatures—for woody evergreen plants.

We visualised the trait distributions in the global climate space defined by three bioclimatic variables for each of the six major FTs (Fig. 3, S8). This allowed for a more intuitive comparison of trait-climate relationships among the FTs and three plant groups. FTs related to the same principal axis of trait variation presented similar distribution patterns in the global climate space. However, the GAMs reveal many more subtle distinctions. For example, among FTs related to the LES, Fig. 3b, e, h shows that LMA in all plant groups increases towards drier climates—but LMA in non-woody plants tends to decrease with growing-season temperature in climates with cold winters (MTCO ≤ −2 °C in Fig. 3b). LMA in woody plants is less sensitive to temperature (Fig. 3e, h); however woody deciduous plants show highest LMA in climates with warmer winters (MTCO ≥ 10 °C in Fig. 3e), while woody evergreen plants show highest LMA in climates with colder winters (MTCO ≤ −2 °C in Fig. 3h). Among FTs representing plant size (Fig. 3c, f, i), H is greatest in climates that are both wet (ln MI > 1) and warm (MGST > 15 °C, MTCO ≥ 10 °C) within all groups.

LA showed a general pattern of increase with moisture within all three plant groups (Fig. 3a, d, g), but LA was less sensitive to moisture in woody plants (Fig. 3d, g) compared to non-woody plants (Fig. 3a; also confirmed by Fig. 1d–f). Among woody evergreens (Fig. 3g), the largest leaves are found in wet climates with warm winters (MTCO > 10 °C) as well as warm summers (MGST > 15 °C). Consistent with their habit, woody deciduous (Fig. 3d) and non-woody plants (Fig. 3a) do not show the same strong sensitivity of LA to cold winters.

## Global maps for the six major traits
For each of the three plant groups, we upscaled the GAMs based on three bioclimatic variables to generate global distribution maps for the six major FTs at 0.1° spatial resolution, separately. The separate trait maps for each plant group are shown in Fig. S9.

## Global maps for all traits
In order to predict global patterns for all 16 FTs having plot-level mean values (CWMs) in the sPlotOpen dataset[38], we fitted new GAMs based on

the same three bioclimatic variables, combined with the (remotely sensed) fractional cover of the three plant groups as additional predictors. All FTs were natural-log transformed prior to the analyses. These GAMs explained up to 77% of the global variation of plot-level mean trait values, with a median adjusted $R^2$ of 52% (average median calculated from 54% and 49%) over all 16 FTs. Best fits ($R_{adj}^2 ≥ 49\%$) were obtained for the following nine FTs: H (77%), SSD (68%), DM (64%), SCD (61%), LMA (57%), LA (56%), LFM (55%), $C_{mass}$ (54%) and $N_{area}$ (49%) (see Table S11 for a summary). We created global trait maps for all 16 FTs based on these GAMs at 0.1° spatial resolution (Fig. 4; see Fig. S12 for all 16 FTs). H, LA, and LFM (Fig. 4a–c) were predicted to be higher in tropical regions and warm and humid environments, with lower values found in cold high-latitude regions and dry temperate environments. The global distribution of LMA (Fig. 4d) might be related to the joint effects of global patterns of LA (Fig. 4b) and LFM (Fig. 4c). Plants with higher LMA were predicted to inhabit tropical regions and drier environments, while plants with lower LMA mainly appeared in humid northern temperate regions. Tropical zones subject to high moisture stress harbour species with higher SSD but lower SCD (Fig. 4e–f). However, plants in humid northern temperate regions and boreal forests had relatively higher values of SCD and lower SSD (Fig. 4e–f). Although FTs within the same trait dimension did not show identical global patterns, their maximum values tended to occur under similar environmental conditions. Like high-LMA plants (Fig. 4d), plants with higher $N_{area}$ (Fig. 4g) were generally favoured in hot and dry environments, supporting earlier findings[51–53]. DM (Fig. 4h) was predicted to be notably high in equatorial regions and warm and humid temperate regions with hot summers, coinciding with H (Fig. 4a). $C_{mass}$ showed little variation across the globe (Fig. 4i). It was found to be slightly lower in deserts and arid temperate regions.

## Comparison to iNaturalist data
Independent global trait maps[41] generated by linking plant observations from the iNaturalist citizen-science project to trait measurements from the TRY database were used as a benchmark for our GAM-based predictions, as the iNaturalist data included all 16 FTs considered in our study. We evaluated the pixel-by-pixel agreement between our GAM-predicted maps with the iNaturalist maps at 2° resolution, chosen because some FTs presented the highest correlations between sPlotOpen and iNaturalist at this spatial scale[41].

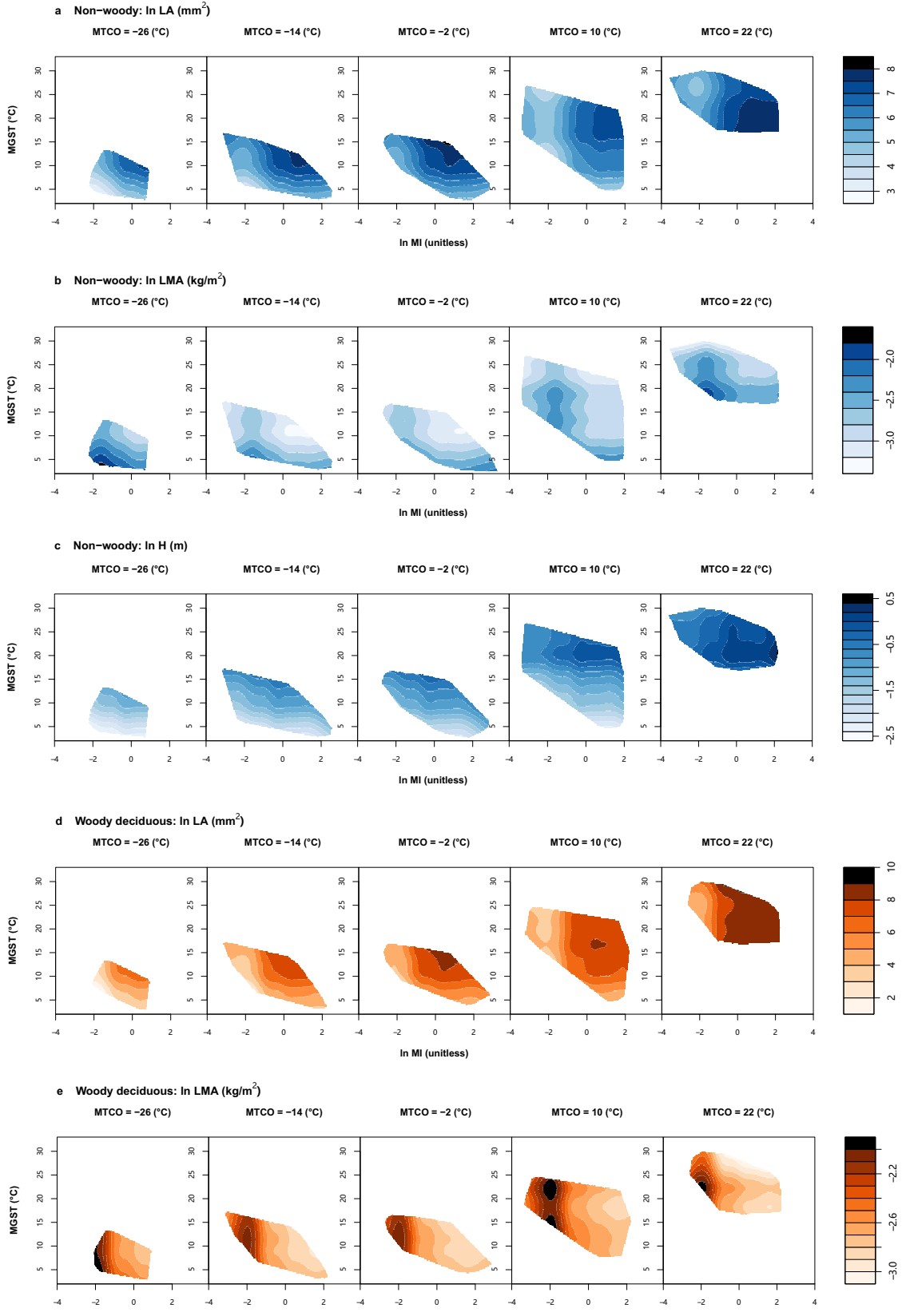

**Fig. 3 | Examples of climate space diagrams.** Climate space diagrams of non-woody (**a**–**c**), woody deciduous (**d**–**f**) and woody evergreen (**g**–**i**) plants: showing distributions of natural-log transformed leaf area (LA, mm²) (**a**, **d**, and **g**), leaf mass per unit area (LMA, kg/m²) (**b**, **e**, and **h**) and plant height (H, m) (**c**, **f**, and **i**) in the global climate space defined by three bioclimatic variables. Fitted trait values are presented as contours, with darker colours in the right colour bar representing higher trait values. Values of ln MI and MGST vary continuously along the horizontal and vertical axes, respectively. Each slice is created at an exact MTCO value. Abbreviations and units of traits are shown in Table 1. ln MI, log-transformed moisture index; MTCO mean temperature of the coldest month, MGST mean growing-season temperature (see Methods for definition). Climate space diagrams for all six major FTs are shown in Fig. S8. All elements of the figure were created by the authors using R v4.2.2 (https://www.r-project.org/)[81].

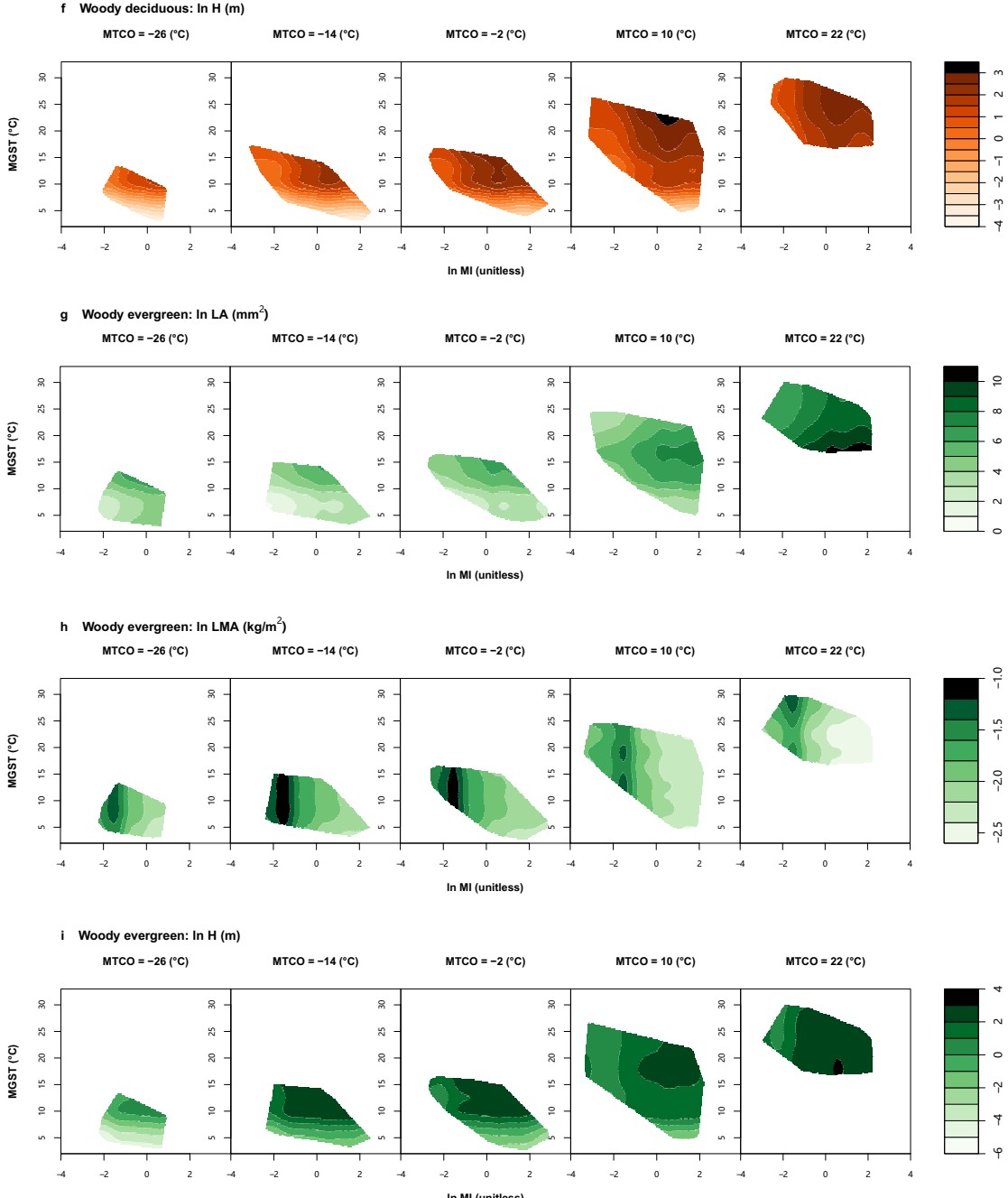

**Fig. 3 |** continued.

The highest $R^2$ and median $R^2$ of linear regressions between our GAM-derived maps and the iNaturalist maps, overall 16 FTs, were 64% and 15%. Figure 5 shows nine FTs that present better agreement than the remaining FTs ($R^2 \geq 13\%$ and slope >0.5 and <2): SCD, LA, LFM, SSD, $\delta^{15}N$, H, $N_{area}$, LMA, and DM (see Table S13 for more information).

The slope of the linear regression also indicates the direction in which the GAM-predicted values deviate from the iNaturalist gridded values. According to Fig. 5, GAMs tended to estimate higher LMA and lower LA and $N_{area}$ in comparison to the iNaturalist data. GAM tended to overestimate SSD at lower values and underestimate SSD at higher values compared with the iNaturalist data. For larger values, GAMs predicted higher H and DM but predicted lower LFM, SCD and $\delta^{15}N$ than the iNaturalist data.

## Discussion
### Ecological strategies reflected by trait-trait and trait-climate relationships

Analysing six major FTs in non-woody, woody deciduous and woody evergreen plants separately, we found that the two orthogonal dimensions of covariation in plant size traits (H–DM) and LES traits (LMA–$N_{area}$), first identified by Díaz and colleagues[12], also apply within each of the three plant groups (Fig. 1a–c). Although Díaz et al.[12] showed that non-woody and woody plants form two almost disjunct hotspots along the H–DM axis[42], our study shows nonetheless that the covariation of H and DM follows almost the same pattern within the non-woody and woody taxon groups (Fig. 1a–c). FTs from the same dimensions of trait covariation tended to be influenced by climate in similar ways (Figs. 1–3 and S8). H relates to light

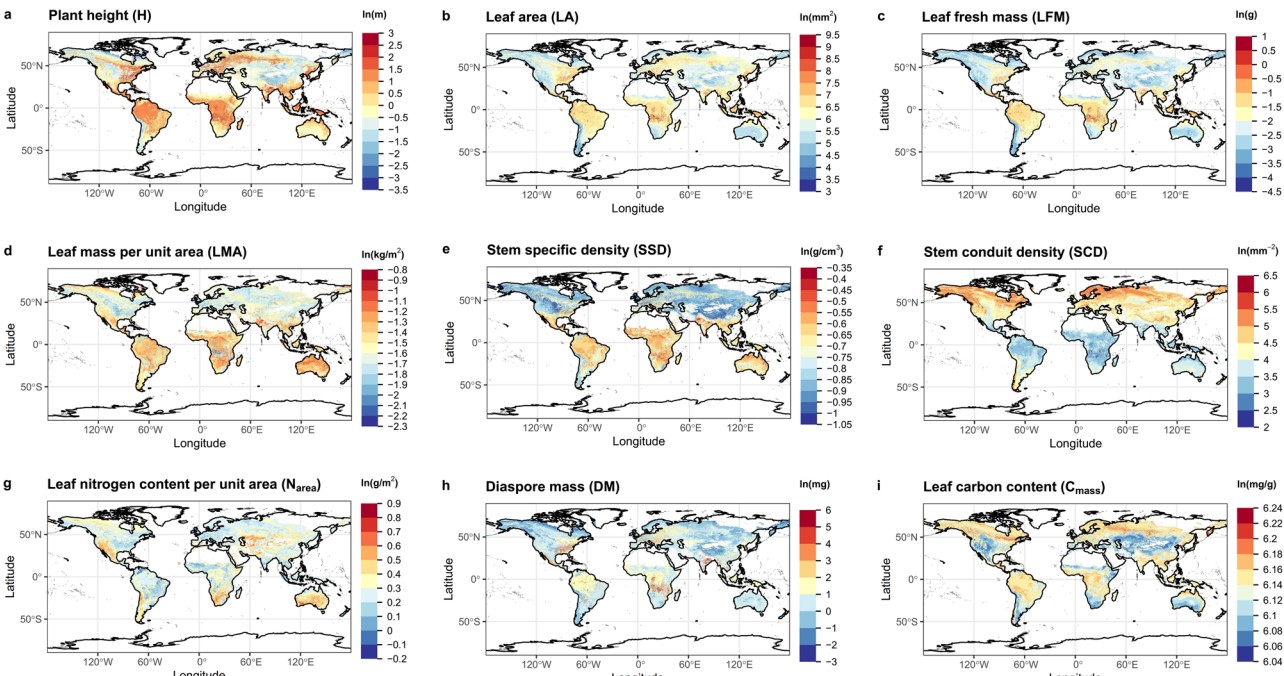

**Fig. 4 | Examples of global trait maps.** Global patterns of plant height (**a**), leaf area (**b**), leaf fresh mass (**c**), leaf mass per unit area (**d**), stem specific density (**e**), stem conduit density (**f**), leaf nitrogen content per unit area (**g**), diaspore mass (**h**) and leaf carbon content (**i**), produced by the best-fit generalised additive models ($R^2 \geq 49\%$) based on three bioclimatic variables combined with remotely sensed fractional cover data for the three plant groups (see Methods for details). The best-fit model here refers to the model with adjusted $R^2$ above the median level of the adjusted $R^2$ over all 16 FTs. Colour gradients from blue to red indicate the increase in trait values. All traits are natural-log transformed. All maps are at 0.1° resolution. Global trait maps of all 16 functional traits are shown in Fig. S12. For maps in GeoTiff format, refer to the Data availability statement. All elements of the figure were created by the authors using R v4.2.2 (https://www.r-project.org/)[81].

competition and DM to dispersal[54,55]. In resource-rich environments (wetter, warmer and more productive sites), plants make more carbon profit that can be invested in stems and also used for seed production. Consequently, they can grow taller to have greater access to light, and produce larger seeds, promoting both a strong competitive status and a high survival probability[2,54–57]. LMA and $N_{area}$ represents resource acquisition strategies[11,12]. LMA reflects a trade-off between the amount of carbon invested in leaves and their lifespan[58]. $N_{area}$ has been considered as the sum of a metabolic component and a structural component, the former proportional to photosynthetic capacity, the latter to LMA[46,47]. Leaves with high LMA and $N_{area}$ are favoured in hot and dry climates because this combination can conserve water to allow the plants to maintain a relatively high photosynthetic rate in the face of increased evapotranspiration[51–53].

Our analysis also revealed a differentiation in trait combinations among the three plant groups, specifically in the covariation of SSD with other FTs. SSD in non-woody plants was strongly correlated with LES traits (Fig. 1a). SSD in woody deciduous plants was relatively independent (Fig. 1b), while SSD in woody evergreen plants was more correlated with plant size traits (Fig. 1c). This novel finding is explicable in adaptive terms. In dry climates, some herbaceous and succulent plants (typically with low SSD) can fix $CO_2$ though their green stems, reducing water loss from the leaves and providing an extra source of carbon[59–61]. The variation of SSD did not show stronger correlation with a single of the two major trait dimensions in the global spectrum of plant form and function[12]. Previous studies were either conducted on global vascular plants[12] or analysed trait data at the community level by using community-weighted means[9,14]. These studies did not identify the functional differences among the non-woody, woody deciduous and woody evergreen plants, consequently failing to detect the distinctive trait combinations associated with SSD. We also extracted pixel values from the GAM-based global maps of the six major traits in each plant group and performed PCA and RDA on them to test these observed trait combinations were still present in the predicted trait values (Fig. S16). The distinct patterns of SSD among the three plant groups were not evident in

the predicted trait values—particularly in non-woody plants, where the previously strong correlation between SSD and LES traits was now absent (Fig. S16a). Trait data used in original PCA and RDA were calculated from the database of Díaz et al.[40] (see Methods) in which observed records for SSD are available only for a few non-woody species; missing values of SSD were imputed via leaf dry matter content (LDMC)[40], which was found to be closely related to both LMA and $N_{area}$[9]: perhaps accounting for the strong correlation between (LDMC-derived) SSD and the LES traits.

Correlations between the six major FTs and the three bioclimatic variables varied among non-woody, woody deciduous and woody evergreen plants. The relative importance of predictors was different among the groups, as were patterns of trait distribution in climate space (Figs. 1–3 and S8). Bioclimatic variables explained more trait variance for woody than non-woody plants (Fig. 1d–f, Table S7). One explanation is that woody plants have permanent organs that must endure climatic conditions year-round over the years. In addition, short-stature non-woody plants are closer to the ground and experience different physical conditions compared with tall-statured woody plants[42]; therefore, the microclimate of non-woody plants in the understorey cannot be directly represented by macroclimate variables[42].

Different trait-climate correlations among the three plant groups are associated with their life-history strategies. For non-woody plants, LMA was less constrained by plant-available moisture (represented by ln MI) than woody plants (Fig. 2), and their DM showed little correlation with all three bioclimatic variables (Figs. 2–3 and S8). Such discrepancies may arise because non-woody species tend to occupy smaller, more specialised and diverse niches than woody species[42,62]. Once climatic conditions are outside their tolerance, they undergo dieback[63]. Therefore, they must grow quickly and reproduce efficiently within the favourable portion of a short lifespan. Such a strategy precludes tall stature[64], supports lower LMA leaves (thus lower leaf $N_{area}$) even in drier climates in order to maintain high photosynthetic and respiration rates[63], and supports the production of copious small seeds to maximise fitness under high resource availability[57,65,66]. The short lifespan of many non-woody plants also represents an adaptive

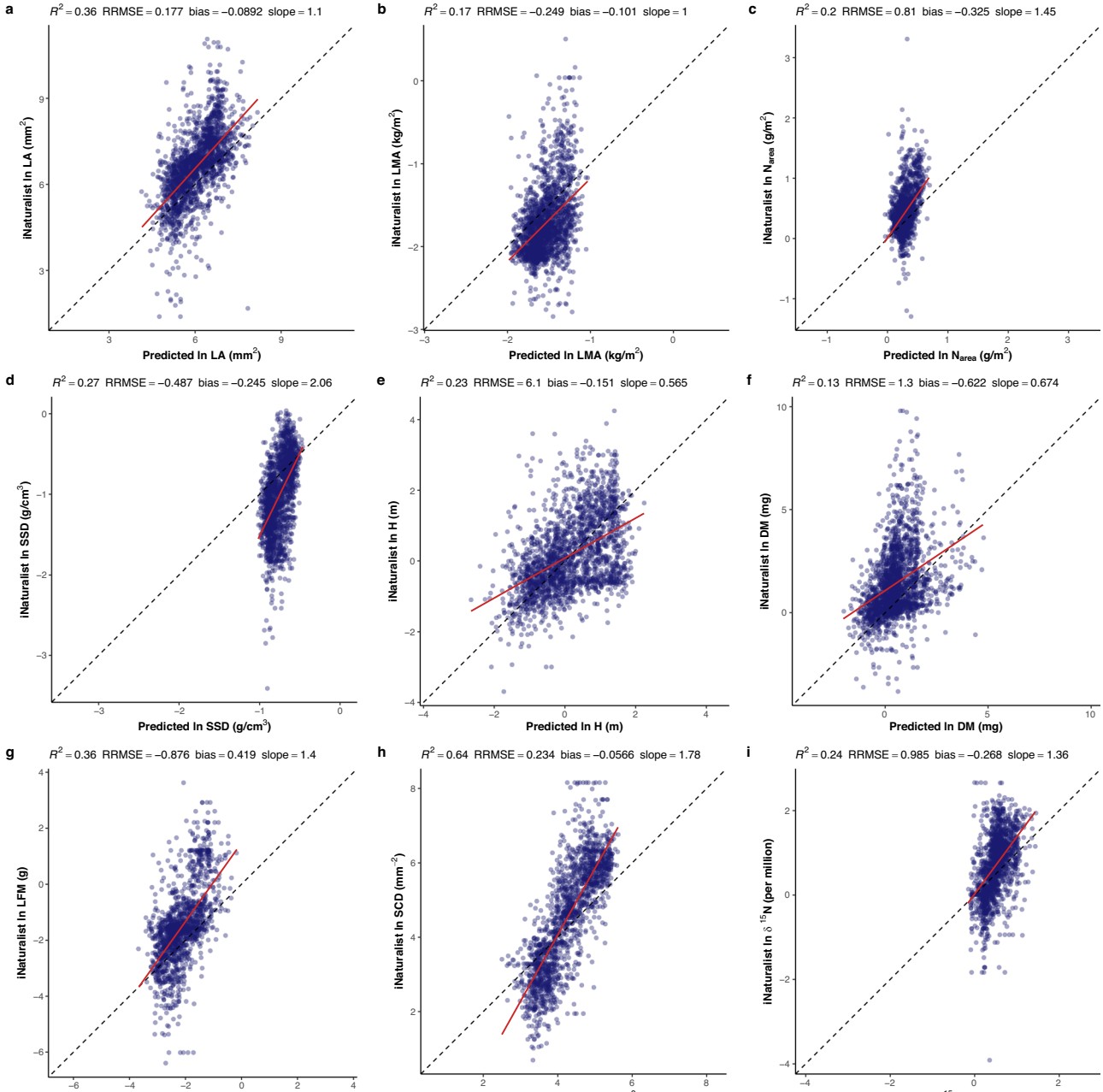

**Fig. 5 | Comparison between predicted trait values and the iNaturalist estimated trait values. a** leaf area (LA); **b** leaf mass per unit area (LMA); **c** leaf nitrogen content per unit area ($N_{area}$); **d**, stem specific density (SSD); **e** plant height (H); **f** diaspore mass (DM); **g** leaf fresh mass (LFM); **h** stem conduit density (SCD); **i** leaf nitrogen isotope ratio ($\delta^{15}N$). Red lines visualise the linear regression between model-predicted map pixel values and the iNaturalist map pixel values estimated at a 2° spatial resolution. The dotted line is the 1:1 line. $R^2$ is the coefficient of determination; RRMSE is the root-mean-square error expressed as a proportion of the observed mean trait value (here observations are represented by iNaturalist trait values); bias is the difference between observed and model-predicted mean values, as a proportion

of the observed mean trait value (here observed values are iNaturalist trait values); slope is the slope of the linear regression of iNaturalist estimations against model predictions. The nine traits displayed here showed higher agreement between model-predicted maps and the iNaturalist maps compared to the remaining traits ($R^2 \geq 13\%$ and slope >0.5 and <2). All traits are natural-log transformed. Scatter plots of model-predicted trait values versus iNaturalist estimated trait values for all 16 traits are shown in Figure S13 and parameters for evaluating linear regressions for all 16 traits are shown in Table S13. Abbreviations and units of traits are shown in Table 1. All elements of the figure were created by the authors using R v4.2.2 (https://www.r-project.org/)[81].

strategy that facilitates their survival in demanding environments. They can live their short lives between fires, physical disturbances, droughts, or annual freezes[64].

Tall woody plants, on the other hand, are successful competitors for light, persisting across years in changing environments[64,65]. However, deciduous and evergreen woody plants also adopt different ecological strategies. We found that LES traits (LMA–$N_{area}$) of woody deciduous plants were insensitive to growing-season temperature (MGST) but were strongly

affected by plant-available moisture (ln MI) and winter temperature (MTCO) (Fig. 3 and S8). This distinction corresponds to the fact that deciduous plants shed their leaves in response to dry or cold conditions[67,68]. As a result, the leaf life cycle of woody deciduous species is tied to the growing season[45]; their lower LMA (and thus lower $N_{area}$) represents an adaptation to a short lifespan[42,45]. In contrast, woody evergreen plants, which need to maintain leaves and vascular networks over successive seasons[64], typically adopt more conservative nutrient-use strategies (higher LMA and

$N_{area}$) and may require greater protection against drought and/or frost damage[11,13,14].

LA in woody plants was positively correlated with winter temperature (MTCO) (Fig. 3), but the relative importance of winter cold for LA in evergreen plants was higher than that in deciduous plants (Fig. 2). Evergreen woody plants with small leaves (low LA) are better adapted to harsh winters. Larger leaves are more prone to frost damage because of their thicker insulating boundary layers of still air that slow sensible heat exchanges with their surroundings[69]. This means that their ability of heat uptake from their surroundings to compensate for the energy lost to the night-time sky is diminished, leaving them vulnerable to frost[69]. Although all three plant groups showed the highest SSD at the same moisture level, the SSD in evergreen plants was higher (Fig. S8) and was less controlled by moisture (Fig. 2) than that of the other two groups. SSD reflects wood density and is related to drought-tolerance of plants[70]. Evergreen trees with higher wood density are less vulnerable to drought-induced embolism than co-existing drought-deciduous trees with lower wood density in arid environments[71]. The hydraulic architecture of high-SSD evergreen woody plants, including fibres and thick-walled vessels, allows them to maintain water transport to the canopy under greater xylem pressure caused by water shortage and to continue growing in seasonally dry climates[70,71].

## Evaluations of global trait maps

Mindful of potential differences in trait-climate correlations among non-woody, woody deciduous and woody evergreen plants, as shown for the six major FTs, we fitted a set of GAMs that included fractional abundances of these groups to simulate global patterns of all 16 FTs. These GAMs predicted up to 77% of variation in our global data sample correctly (Table S11). We evaluated the agreement of our GAM-predicted maps with the iNaturalist maps[41]. The iNaturalist[41] data represents a large and independent trait dataset based on citizen-science plant observations and trait measurements from the TRY database[39]. The data included all 16 FTs considered in our study. Previously published global trait maps generated by statistical modelling or machine learning methods[2,3,25,29,31–34] have only simulated global patterns for a subset of FTs we studied here; moreover, these maps show large differences in data sources, methodology and results[35,36]. We also compared the general patterns of global trait maps generated herein with those from other published products for six major traits and LDMC. The predictability of our GAMs for the common six major traits was significantly stronger than models of other products (Table S12). Almost all six major traits presented similar global spatial patterns to that of previous studies. Nevertheless, regional differences emerged between our results and other products. For example, $N_{area}$ values in Central Africa were at intermediate level in our GAM-predicted map (Fig. 4g), contrasting with the relatively lowest values observed in the theory-based map by Dong et al.[35]; plant height (H) was predicted to be higher in Europe in this study compared to Schiller et al.'s predictions[33]. The predictability of GAM of this study for LDMC was relatively weaker ($R_{adj}^2 = 0.39$, Table S12) and the global pattern of LDMC simulated in this study suggested significant deviation from the previous remote-sensing-based projections[29].

Overall, our GAM-predicted maps showed variably strong correlations with the iNaturalist maps, with the highest $R^2$ of 64% but a median $R^2$ of only 15% (Table S13). They showed varying degrees of overestimation or underestimation compared to the iNaturalist maps (Fig. S13). We first checked which regions across the globe showed overestimation and underestimation of our predicted FTs compared to iNaturalist data (Fig. S14). Our GAMs tended to predict higher LMA, DM and SSD and lower SCD than iNaturalist data in the northern temperate regions, but lower $N_{area}$, SSD and LFM than iNaturalist data in southern tropical and temperate regions. H was overestimated globally by the GAM compared to iNaturalist, especially in the northern high latitudes.

Plot-level mean values of 16 FTs in our study were obtained from the sPlotOpen dataset. In principle, the above disagreements may arise from the respective sampling biases of iNaturalist and sPlotOpen. The iNaturalist contributors are by no means evenly sampled across growth forms (tree/ shrub/herb)[41], while the sPlotOpen data cover a relatively small portion of the global climate space. The coverage of growth form in each grid cell of the iNaturalist maps indicated a bias towards herbaceous plants in the northern regions, and towards woody plants further south. This is because plant observations in iNaturalist are largely governed by individual decisions[33,41] and are a priori likely to underrepresent grasses[41], for example. The data are nonetheless of value because of their independence from other data sources. The sPlotOpen dataset has both commonalities and disagreements with the iNaturalist data[41]. sPlotOpen is an aggregation of vegetation plots from various studies with specific and distinct research aims across global eco-systems that do not share the same criteria in the sampling approach, which may also lead to biases in this dataset[37,38]. Large areas of the globe, including Canada, Central South America, Central Africa, Central Asia, East Asia, and Siberia, are undersampled (Fig. S4). This sampling bias affects the robust-ness of prediction of trait distribution for large areas. Furthermore, while sPlotOpen provides plot-level means for each trait, not all species within every vegetation plot have been recorded. 16.5% of plots have all vascular plant records, plant records of 72.1% plots are not specified, and the remaining 11.4% of plots have only certain species records (e.g., only woody plants or dominant plants) in the final plot-level mean trait dataset used in this study (Table S2). As a result, plot-level trait means sampled for specific target plant in a vegetation plot cannot accurately reflect the CWMs of traits within the corresponding regions.

The selection of predictor variables in GAMs might contribute to the disagreement between the two set of global trait maps. Our study applied a consistent set of three bioclimatic variables to each FT, thereby preserving model simplicity and providing a consistent baseline for comparison. The three selected variables reflect three most important aspects of climate that govern plant distributions, i.e., winter cold, summer warmth and moisture availability[48,49,50]. These three variables effectively explained global patterns of different FTs and sufficiently distinguished the main differences in trait-climate relationships among three plant groups. However, some FTs can be influenced by other factors in addition to the bioclimatic variables used in this study. Soil fertility factors and soil age have been indicated to correlate with leaf nutrient traits[3,14,72–74]. Land use and disturbance regimes can also affect trait-environment relationships[25,32,75]. Disturbance by fire or grazing and browsing by megaherbivores have strong effects on traits of seed dispersal and plant regeneration[25]. Atmospheric $CO_2$ concentration is also an important driver of some FTs, directly affecting the growth rates and pro-ductivity of plants[25,76]. Evolutionary history effects (represented by phylo-geny) were reported to explain on average more than two-thirds of the variability of the foliar concentrations of N, P and K[3].

The scale at which trait variation occurs may be another reason for the lower correlations between GAM maps and the iNaturalist data for some FTs. Some FTs lack macroecological patterns, such as $C_{mass}$, which showed little variation on a global scale (Fig. 4i). In addition, our community-level prediction ignores intraspecific trait variation. A global meta-analysis based on 36 plant traits indicated that traits can vary sub-stantially among individuals within species, and that the relative amount of intraspecific trait variation does not vary with plant growth forms or climate[77].

This study has characterised universal relationships among FTs and between FTs and climates within non-woody, woody deciduous and woody evergreen plants, and simulated global patterns of FTs based on these relationships. It has highlighted certain key differences between the three plant groups in their ecological strategies as represented by trait-climate correlations, indicating that it is useful to differentiate between non-woody, woody deciduous and woody evergreen plants in large-scale, trait-based studies. Based on three bioclimatic variables and global vegetation cover and their interaction effects, we can explain up to three-quarters (on average about a half) of global variation of community-weighted means for all 16 FTs. By generating global maps for all 16 FTs at 0.1° resolution, we have provided the most comprehensive set of trait maps based on statistical upscaling approach to date.

## Methods

### Plant functional traits and three plant groups

The FT data were collected from the sPlotOpen dataset[38] and the enhanced species-level trait dataset of Díaz et al.[40] (see Table 1).

The sPlotOpen dataset provides plot-level trait means (as community-weighted means, CWMs) of 18 FTs, as well as plant taxa within each vegetation plot and their relative abundances[38]. We calculated the LMA and area-based leaf phosphorus content ($P_{area}$) from specific leaf area (SLA) and mass-based leaf phosphorus content ($P_{mass}$) in the sPlotOpen dataset according to Eqs. (1) and (2). After removing some redundant traits, such as SLA, $P_{mass}$, mass-based leaf nitrogen content ($N_{mass}$) and leaf N:P ratio (which can be directly calculated from $N_{area}$ and $P_{area}$), 16 FTs were selected for analysis (Table 1). All 16 FTs were natural-log transformed before analysis. Here we focus on $N_{area}$ and $P_{area}$, because Osnas et al.[78] showed that leaf nitrogen and phosphorus contents are approximately distributed proportional to leaf area rather than mass. $N_{area}$ and $P_{area}$ were found to be proportional to LMA and photosynthetic capacity of plants[46,47,79]. Both LMA and photosynthetic capacity are quantitatively predictable from climate[45,79], which may lead to potentially high predictability of $N_{area}$ and $P_{area}$ from climates. In contrast, values of $N_{mass}$ and $P_{mass}$ may be more conservative under climate change. We also conducted the following analyses on $N_{mass}$ and $P_{mass}$ (see Supplementary Information SI 2 for details) and found that climate is not good predictor of their distributions.

$$LMA\ (kg/m^2) = \frac{1}{SLA\ (m^2/kg)} \tag{1}$$

$$P_{area}\ (g/m^2) = \frac{P_{mass}\ (mg/g)}{SLA\ (m^2/kg)} \tag{2}$$

All plant taxa within each vegetation plot in the sPlotOpen data were extracted. We collected plant information including woodiness, plant growth form and leaf phenology for each taxon from public databases, regional floras, and published papers (see Data availability for relevant references of plant information). Each plant taxon was then assigned to a growth form according to the collected plant information to create a plant growth form dataset. There are 42,676 unique taxa in the dataset. Table S3 presents 16 growth forms in total and the proportion of each growth form in the plant growth form dataset.

We classified all taxa as non-woody, woody deciduous or woody evergreen according to their growth forms. Some minority growth forms, including ferns, palms, cycads, bamboos, cacti and succulents, were omitted from subsequent analyses. We then used the relative coverage of each taxon within a vegetation plot provided by the sPlotOpen data and the assigned probability of each taxon being in each plant group (Table S4) to calculate the relative coverage of each of the three plant groups within a plot. The calculation followed Eqs. (3) and (4).

$$RA\ (group) = \sum \left(RC(taxon) \times P\ (group)\right) \tag{3}$$

$$RC\ (group) = \frac{RA\ (group)}{\sum \left(RA\ (group)\right)} \tag{4}$$

where RA(group) is the relative abundance of each plant group within a vegetation plot; RC(taxon) is the relative coverage of each taxon within a vegetation plot provided by the sPlotOpen dataset; P(group) is the probability of each taxon being in each plant group, which is derived from the probability of each growth form being in each plant group; RC(group) is the relative coverage of each plant group within a vegetation plot.

The species-level trait dataset from Díaz et al.[40] provides species-mean values of six major FTs that constitute the global spectrum of plant form and function[12]. We converted $N_{mass}$ to area-based leaf nitrogen content ($N_{area}$) based on the raw data, according to the Eq. (5). After merging this species-level trait dataset with the sPlotOpen dataset and the

plant growth form dataset, a total of 35,773 common species were explicitly classified into non-woody, woody deciduous and woody evergreen groups. The scientific names of all taxa from different datasets were aligned before integration by using Taxonomic Name Resolution Service V5.2 (TNRS, https://tnrs.biendata.org/)[80]. We then calculated CWMs for the six major FTs of each of the three plant groups within each vegetation plot based on their species-level means and the relative coverage of each plant group. These CWMs were then natural-log transformed prior to further analyses. All data manipulations and following analyses were conducted in R v4.2.2 (https://www.r-project.org/)[81].

$$N_{area}\ (g/m^2) = \frac{N_{mass}\ (mg/g)}{SLA\ (m^2/kg)} \tag{5}$$

### Bioclimatic variables

Three bioclimatic variables were selected[48,49,50] (Fig. S2): (i) moisture index (MI, unitless) representing plant-available moisture—the natural-log transformed MI (ln MI) was used in the analysis to emphasise differences at the dry end of the moisture range[82]; (ii) mean temperature of the coldest month (MTCO, °C) representing winter cold; (iii) mean growing-season temperature (MGST, °C) representing summer warmth. The three selected variables correspond to the three recognised climatic dimensions that regulate global geographic distribution of vegetation[48,49,50]. They control the plant distribution limits by affecting the plant attributes that determine physiological processes and adaptive strategies, such as life form, leaf phenology, leaf size and stomatal conductance[48,49,50]. Hence, the distribution of some key plant traits is likely to be a function of these bioclimatic variables, and thus may be quantitatively predicted by these variables[48].

We collected gridded climatological data for further calculations, with a spatial resolution of 30 arc-second (ca 0.01°). Growing degree days (heat sum) above 0 °C ($GDD_0$, °C), number of growing degree days above a base level of 0 °C ($NGD_0$, number of days), monthly averaged mean daily air temperature (tas, K/10) and mean monthly precipitation amount (pr, kg m$^{-2}$ month$^{-1/100}$) were obtained from CHELSA V2.1 (https://chelsa-climate.org/)[83,84], covering the period from 1981 to 2010. Downward total shortwave solar radiation data accumulated over the month (srad, kJ m$^{-2}$) were downloaded from the CHELSA V1.2 (https://chelsa-climate.org/)[83,84], covering the period from 1979 to 2013. Global elevation derived from the SRTM elevation data (elev, m) was acquired from WorldClim v.2.1 (https://www.worldclim.org/)[85].

We used the Simple Process-Led Algorithms for Simulating Habitats (SPLASH) model[86,87] with climatic variables as inputs to calculate the mean monthly potential evapotranspiration (PET, mm). We extracted the climatic values for each vegetation plot according to its longitude and latitude. The 'splash.point()' function in R *rsplash* package[86,87] was applied to calculate the PET of each plot. We then decreased the resolution of raw climatic raster data from 0.01° to 0.1° and fitted them into the SPLASH model by 'splash.grid()' function in R *rsplash* package[86,87] to calculate the global PET. Three bioclimatic variables (Fig. S2) were then calculated as follows:

$$MI\ (unitless) = \frac{\sum_{k=1}^{12} pr_k}{\sum_{k=1}^{12} PET_k} \tag{6}$$

$$MTCO\ (°C) = \min_{1 \le k \le 12} tas_k \tag{7}$$

$$MGST\ (°C) = \frac{GDD_0}{NGD_0} \tag{8}$$

where $pr_k$ (mm) is mean monthly precipitation; $PET_k$ (mm) is mean monthly potential evapotranspiration calculated from the SPLASH model; $tas_k$ (°C) is monthly average temperature; $GDD_0$ (°C) are growing degree days heat sum above 0 °C; and $NGD_0$ (number of days) is number of growing degree days above a base level of 0 °C.

**Vegetation coverage**

We checked the 'naturalness' of each vegetation plot in our trait dataset against the global map of land cover provided by the ESA CCI-LC database (https://www.esa-landcover-cci.org/?q=node/164)[88] at 300 m spatial resolution in 2010. For plots belonging to all types of croplands, urban areas, bare areas, water bodies and permanent snow and ice in the ESA CCI-LC database, we defined them as 'unnatural vegetation' and removed those plots from the trait dataset used for data analysis. After matching the trait dataset with bioclimate data, 77,074 natural vegetation plots remained in the final dataset for our study.

The ESA CCI-LC global land cover data were also used to calculate the global fractional cover of three plant groups. The spatial resolution of the raw land cover raster was decreased from 300 m to 0.1° to match with the resolution of bioclimate raster for further analyses. This ESA CCI-LC dataset contains 23 classes including 38 sub-classes of global land cover (Table S5). We first selected the vegetation plots that contained records of 'all vascular plants' in our trait dataset (12,743 plots in total). We then identified the land cover class and sub-class of each of these vegetation plots according to the ESA CCI-LC maps at original 300 m resolution and accordingly calculated the median relative cover of the three plant groups for each sub-class. The coverages of three plant groups of artificial ecosystems and non-natural and non-terrestrial vegetations were set to be NA. We then replaced grid values representing sub-classes of global land cover in the ESA CCI-LC maps with the median relative cover of the three plant groups within each sub-class and calculated the average values at 0.1° resolution to produce the global fractional cover of the three plant groups.

**Multivariate analysis**

The natural-log transformed CWMs of six major FTs of each plant group were rescaled to a mean of 0 and a standard deviation of 1 and then subjected to a PCA and a Redundancy Analysis (RDA), conducted by 'prcomp()' function and 'rda()' function in the R *vegan* package[89], respectively. PCA was used to evaluate the covariation of trait combinations among six FTs in different plant groups. RDA was applied to describe the extent to which the variation of trait combinations (response variables) can be explained by bioclimatic variables (explanatory variables).

**Generalised additive models**

Correlations between FTs and three bioclimatic variables were characterised and visualised by GAMs. GAMs were fitted for natural-log transformed CWMs of six major FTs in each of the three plant groups with three bioclimatic variables as explanatory variables, without interaction terms, using the 'gam()' function in R *mgcv* package[90]. We used the RMEL (restricted maximum likelihood) as smoothing parameter to control the smoothness of the predictive functions. We then calculated the relative importance of each bioclimatic variable in the GAMs to determine which of the predictors is more significantly related to the distribution of each FT. The concurvity of three bioclimatic variables in GAMs was checked before fitting models (mean values: ln MI = 0.487 ± 0.06, MTCO = 0.651 ± 0.07, MGST = 0.867 ± 0.02). MGST presented relatively high concurvity, which makes it challenging to interpret the individual importance of variables in trait prediction. Nonetheless, the overall predictive performance of the GAMs remains robust, with the combined effect of the predictors effectively capturing the variation in FTs.

We sampled the bioclimates at regular intervals from minimum to maximum over the globe, e.g., sampling at increments of 0.05 (unitless) for ln MI, 0.25 °C for MTCO, and 0.25 °C for MGST, to generate three sample datasets simulating the global climate space. The sample data and GAMs were then used to predict the distribution of trait values in the global climate space for each plant group. The response surfaces of trait values (natural-log transformed) resulting from GAMs were presented as contours in the three-variable climate space in the form of two-dimensional slices. Convex hulls generated by 'convhulln()' function in the R *geometry* package[91] were used to avoid representing parts of the fitted surface that are not constrained by data. For each plant group separately, we also predicted and mapped the global distribution of natural-log transformed values of six major FTs, as predicted by the GAMs, based on global raster data for the three bioclimatic variables at 0.1° spatial resolution.

**Global trait maps and evaluation**

In order to simulate global trait patterns for all plant taxa, we fitted new GAMs for all 16 FTs with natural-log transformed plot-level means (CWMs). In the new GAMs, trait values were predicted by not only three bioclimatic variables, but also global fractional cover of plant groups, and interactions between each bioclimatic variable with the cover fraction of each plant group. As the sum of the global fractional cover for the three plant groups is 1, we only considered the coverage of two of these groups (woody deciduous and woody evergreen) when fitting the new GAMs. The new GAMs also used the RMEL as smoothing parameter and were fitted using the 'gam()' function in the R *mgcv* package[90]. We then generated global trait maps for all 16 FTs at 0.1° spatial resolution.

Although using a weighted average of three maps from different plant groups should be a more straightforward way to generate global maps, we did not adopt this approach. This is because simulating separate global trait patterns for each of the three plant groups and calculating weighted averages of them with fractional cover of plant groups as weights would only be applicable where traits possess species-level values. In this study, only the six major traits are provided with species-level mean values. The remaining traits have plot-level mean values, which are weighted averages of trait values for all plant taxa within the plot. It is almost impossible to predict global trait patterns for each plant group from such community-weighted mean trait values. However, using the GAMs, including fractional covers of plant groups, provides a more universal alternative approach, which is directly applicable to plot-level means.

We tested our global trait maps based on GAMs predictions against another set of global trait maps[41] based on the combinations of citizen-science plant observations from the iNaturalist project and trait measurements from the TRY database. The iNaturalist-based maps provided all FTs available for the calculation of the same 16 FTs as in our study. The plot-level means of all 16 FTs in our study were taken from the sPlotOpen, and the global trait patterns estimated by the iNaturalist tend to most strongly resemble that of the sPlotOpen at 2° resolution[41]. Therefore, we aggregated and reprojected our global trait maps according to the iNaturalist maps (2° spatial resolution) using bilinear interpolation by 'projectRaster()' function in the R *raster* package[92]. Then, we assessed the pixel-by-pixel agreement between our trait maps and the iNaturalist trait maps by linear regression using the 'lm()' function in the R *stats* package.

**Statistics and reproducibility**

All of the above statistical analyses are reproducible by following the procedures in Methods. All data and R scripts for carrying out all the above analyses are provided in the Data availability and Code availability sections.

**Data availability**

The plant functional traits are collected from openly accessible databases: the sPlotOpen dataset[38] and the enhanced species-level trait dataset of Díaz et al.[40]. Raw climate data are obtained from CHELSA V2.1 (https://chelsa-climate.org/)[83,84]. Global land cover maps are from the ESA CCI-LC database (https://www.esa-landcover-cci.org/?q=node/164)[88]. Plant growth form dataset, global maps of three bioclimatic variables (GeoTiff format), global trait maps for 16 plant functional traits and separate maps of 6 major traits for non-woody, woody deciduous and evergreen plants (GeoTiff format), global fractional cover of non-woody, woody deciduous and woody evergreen plants (GeoTiff format), as well as compiled datasets used for all analyses in this study are openly available at: https://doi.org/10.5281/zenodo.13325275 (ref. 93).

**Code availability**

The reproducible R Scripts used to conduct all data manipulations and analyses are available at: https://doi.org/10.5281/zenodo.13325275 (ref. 93).

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

## Acknowledgements

J.L. is funded by the scholarship granted by the China Scholarship Council (CSC No. 202108060263). I.C.P. acknowledges funding from the European Research Council (ERC) under the European Union's Horizon 2020 research and innovation programme (grant no. 787203 REALM). I.C.P. also acknowledges support from the Land Ecosystem Models based On New

Theory, obseRvations and ExperimEnts (LEMONTREE) project funded through the generosity of Eric and Wendy Schmidt by recommendation of the Schmidt Futures programme. J.L. and I.C.P. also acknowledge funding from the Imperial College London Open Access Fund.

## Author contributions

J.L. and I.C.P. conceived the project and developed the methods. J.L. carried out all analyses and drafted the manuscript. J.L. and I.C.P. revised the manuscript.

## Competing interests

The authors declare no competing interests.
