## [Transparent Peer Review file · Communications Biology]

Global patterns of plant functional traits and their relationships to climate

Corresponding Author: Ms Jiaze Li

Figures originally included in the author's rebuttal have been redacted from this file.

Version 0:

Reviewer comments:

Reviewer #1

(Remarks to the Author)
BRIEF SUMMARY

The present work aims to generate and test global maps for 16 important plant functional traits. For this purpose, generally available global data from sPlotOpen (vegetation plots with species abundances and community weighted traits), from the "global spectrum of plant form and function" (compiled traits at species level), and iNaturalist (species occurrences and compiled trait maps) are used along with global climate (CHELSA, Worldclim) and land cover data (ESA CCI-LC). For six major traits maps are also modeled separately for the three groups of non-woody, woody deciduous and woody evergreen plants. The maps are finally compared with independent traits values based on iNaturalist (and TRY), confirming the general spatial trait patterns.

OVERALL IMPRESSION

Although global maps of individual plant traits already exist, the present work is comprehensive and additionally distinguishes three different growth forms. On the other hand, and in addition to the more specific comments below, I missed (1) a clear overview of the workflow including the directly used vs. recalculated traits, major traits vs. other traits, trait maps for plant groups vs. all plants, (2) a more detailed justification of the GAM variable selection for the global predictions, and (3) a discussion of the limitations of this work. Overall, however, I believe that a revised version can make a stimulating and valuable contribution to the current discussion on global patterns of plant functional traits, their drivers, and possible future dynamics.

SPECIFIC COMMENTS

- page 3: I missed the information about previous approaches to global maps of similar traits (see comment in page 17 below).

- page 4: The number of species is impressive, but not all traits are available for all species. An extended Table 1 in the Appendix would be helpful - with information on the number of species per trait with available trait values and the origin of the pixel values used as response for the GAM calibration and evaluation (directly from sPlotOpen or iNaturalist publications vs. calculated for this study).

- page 5, Table 1 - just as a remark: Although sufficiently justified in the manuscript, it is still a pity that N.area and not N.mass was used. This makes a direct comparison with Diaz et al. 2016 more difficult.

- page 7, Fig. 1: Because the orientation of PCA and RDA axes is arbitrary, please check to flip the axes of each plot to show in all plots the same axes orientation. For example, the two axes of plot b and c point in opposite directions. To facilitate the comparison, Fig. 2 of Diaz et al. 2016 might be used as reference.

- page 7: The use of only three bioclimatic variables and the same variables for all FTs can be questioned and requires some justification. Without question, the three bioclimatic variables have important impacts on plants. However, this does not necessarily apply to individual traits, which most likely differ in their bioclimatic drivers.

- page 8, GAM calibration: Temperature related bioclimatic variables are often highly correlated. This seems also the case for MGST and MTCO as indicated by the RDA plots in Figure 1. Did you check for collinearity and how did you deal with it?

- page 9, Fig. 3: I was confused because "MTCO < 4°C" is not visible in Fig. 3 and it was also unclear what the MTCO values like "MTCO = -2 (°C)" mean - probably not an exact value, but a range of values? Fig. 3 is also very small to read. It might be better to focus on individual diagrams as examples or to move the entire figure to the Appendix.

- pages 11 and 23: The GAMs for the global maps included the fractional cover of two (not three) plant groups as additional predictors as well as their interaction terms with bioclimatic predictors. Although this seems reasonable, wouldn't a simple weighted average of the three maps of all three plant groups using fractional cover as weights be more straightforward? Did you test such simple GAM ensembles (i.e., the weighted average of the three maps per trait in Fig. S8)?

- page 12, Fig. 4: These maps are very small again. A 2x2 or 2x3 layout with only two columns seems more reader-friendly.

- page 17: "There are also previously published global trait maps generated from statistical modeling or machine learning methods". I missed this information in the introduction. In addition, remote sensing was used in previous studies. There is a recent publication with similar maps for leaf traits by the same last author: Dong et al. 2022, GEB, DOI: 10.1111/geb.13680. There, the overall global pattern for N.mass as an example is confirmed, but a comparison also shows some regional differences. A comparison with already published global trait maps would clearly improve the discussion section.

- page 19: "We collected plant information including woodiness, plant growth form and leaf phenology for each taxon from regional floras, public databases and published papers." It remains unclear where the information mainly derived from. Because regional floras are mentioned first, it looks like the most important source (and a big effort!), but the Diaz et al. 2022 already includes woodiness, growth form, and leaf type.

- methods: A short comment on the taxonomic and nomenclature backbone of the various sources would be helpful. Were the raw data already harmonized?

- discussion of limitations: Besides the partly very (too?) detailed discussion about the trait-trait and trait-climate relationships and about the iNaturalist evaluation, a discussion about limitations of the derived maps is missing. As stated in the sPlotOpen publication, these data "comes with a number of warnings" - first of all the sampling bias (as clearly shown in Fig. S2 of this MS). Also the robustness of the global trait patterns remains unclear (e.g. when using a different variable selection for GAMs or a simple ensemble approach as mentioned above). Already published global traits maps also show some differences worth to discuss (see comment on page 17 above). And the plant groups of this MS are still very heterogeneous - e.g. in respect to life span (annuals vs. perennials) or evolutionary history (gymnosperms vs. angiosperms). On the other hand, the comprehensive trait maps of this submission will provide a valuable data source for global studies in vegetation ecology and beyond.

Reviewer #2

(Remarks to the Author)

General comments

It was very interesting to read the article titled 'Global Patterns of Plant Functional Traits and Their Relationships to Climate.' Despite there being more studies modelling the spatial distribution of traits at a continental and global scale, as the authors pointed out, I find the authors' analysis across different plant groups (e.g., woody and non-woody) highly valuable. However, there are several points throughout the paper that require substantial improvement.

Generally, several parts of the introduction lack the necessary ecological context for the concepts the authors mention (see detail recommendation in line by line comments). Furthermore, key methods and data deserve a concise yet improved explanation in the main text. For instance, it's challenging for readers to determine from where the climatic variables and remote sensing data used for analysis were obtained. Currently, readers might get the impression that the authors directly 'calculated' these variables, which is not the case. I also recommend improving the figures, their titles and captions to ensure that readers can readily grasp the key takeaways and main messages of the figures (see line by line comments).

The discussion could be streamlined. In some parts the authors should focus more on the meaning of the results, that is, avoid redundant findings or explanations of functional ecology concepts. Instead, contrast more your results with the ecological concepts and ideas.

Because 'global' studies can be controversial, I strongly suggest the authors to briefly address at the end the discussion limitations of the data used and its impact on the results. For instance, vast portions of the globe, such as the north of Africa (Fig. S2), lack any available plot. This affects the robustness of prediction of trait distribution for large areas.

It is amazing that the authors followed the key principle of the scientific method by sharing their code and data to make their study reproducible. But the authors need to revise the data related to the code because I could not find the files required to actually replicate the study (i.e., text, .xlsx., csv).

Lastly, I strongly advise minimizing the excessive use of abbreviations throughout the text, as it can make the document difficult to read. While I understand that certain abbreviations are essential, I advise to find a balance to prevent an overload of them. I also recommend avoid using passive voice when describing methods in the results section (see line by line comments).

Line by line comments

L 10-11: mention at least one strategy or adaptation that relate to ecosystems and their composition. Otherwise the starting of the abstract reads very vague.

L 18: what do the authors mean by traits being higher than others? Do you mean higher values, perhaps larger leaf areas or more acquisitive? A clarification is needed.

L 27: Provide at least one trait that relates to the process you are mention afterwards, i.e., life history or responses to environmental change, to make the paragraph less vague.

L 42: Explain what kind of function you are referring to, to avoid the text being vague.

L 43-44. Please specify what functions are meant here. It is not enough to say that quantifying is important. Also, it is not clear what 'their controls' mean.

L 48: briefly exemplify a type of shift.

L 100: Use active voice.

L 108-110: this is a very interesting result!

L 112: (Generally Figure 2) Figures should be able to stand alone. Include the abbreviation meanings in the figure caption, else readers won't understand the meaning of them. Remove the 's' in the word 'components'. Incorporate titles (non-woody, woody deciduous, etc) into the figure to make it easier to follow the results, see for example Figure 2 in Barajas Barbosa et al 2023 <https://www.nature.com/articles/s41586-023-06305-z/figures/2>

L 122: add the citation for this data so that readers can know the source of extracted the climatic variables.

L 126: use active voice.

L 131: instead of using the abbreviation, the text would be easier to read by mentioning the traits, meaning height and diaspore mass.

L 137-139: it is unclear to which figure on the main text this analysis corresponds to. Use active voice here as well.

L 139-140: Complete the sentence and improve the explanation of why did you do this analysis. I guess you want to say that the relative importance of a climatic variable in relation to a trait, which is what you show in Figure 2.

L 149: Again, do not use the abbreviation in the figure. You can use the whole name of the variable to make the figures more impactful. E.g., instead of using lnMI, write Moisture index. In the caption you can clarify that you log-transformed.

L 154-155: please explain why are you doing this analysis and what is the key message of Figure 3?

165: improve the readability of figure 3. Add distinctions between the panels and use them here to guide the reader to where exactly this result is found on the figure.

L 172: (Generally Figure 3) It is hard to understand to which analysis the figure corresponds to, I assume from the GAM, but this should be crystal clear for readers. The figure is generally very hard to follow, as it has many panels. You can add distinctions for the figure, e.g., for trait LMA add d, e and f panels, and for trait H add g, h and i panels. Use these distinctions to link the figure in the text. Explain what is the meaning of the legend on the right side of the spaces.

L 178: Explain exactly what remote sensed data/product are you using and cite the reference of origin of the data.

L 206: (Generally Figure 4) I find this figure is the one that best matches your title and the one that readers are expecting to see. But the figure title is not informative yet. Try to improve the title by saying, e.g., 'Global distribution of nine plant traits. Colours indicate increase in x trait value...etc..'. Indicate the meaning of the legend on the right side of each map. Give more relevance to the traits in the maps by avoiding abbreviations on the figure, there is enough space for you to write the name each trait on top of each map. Only mention in the caption that traits were log transform. Try to improve the aesthetics, i.e., plot a 3x3 panel figure and slightly increase the map sizes.

L 213: Explain to the readers why is it a bench mark and how the comparison will improve the message of your study.

L 252: What does it mean 'same trait dimension'?

264-265: This is a very short part where you nicely link your results and explain the meaning of them. But the rest of the paragraph predominantly outlines existing knowledge with little linking to the contribution of your results.

L 277-281: This is a nice explanation to the readers of your results!

Reviewer #3

(Remarks to the Author)

I applaud the focus, trying to predict coordinated sets of FTs rather than isolated ones, and trying to get a much better handle than before on three major growth forms. The analytical methods are sound, the graphs informative and the datasets used the best available in terms of traits and plots. The use of iNaturalist information as validation is clever and fresh.

And by focusing on co-variation of traits, and how they are affected by climate and growth form, this study in parts touches upon a key issue: traits are coordinated together in whole plants, and it is the combination what has to be viable.

So I don't have major doubts or criticisms about the general focus, methods or results. I do, however, wonder about the fundamental advancement of the field that this article represents, which would merit publication in *Comms Biol* and not a good specialized journal. At the moment, I cannot see that very clearly. The concluding statements that this study shows that "it is useful to differentiate between non-woody, woody deciduous and woody evergreen plants in large-scale, trait-based studies" and "By generating global maps for all 16 FTs at 0.1° resolution, we have provided the most comprehensive set of trait maps based on statistical upscaling approach to date" do not seem to be enough. The different alternative explanations for some of the patterns found, while candid, are a bit vague and unsatisfactory. And one of the potentially most novel aspects of the study, the coordination of traits into successful (or not) combinations depending on climate and growth form, is just superficially, almost implicitly touched upon. This limitation might just be the result of not effective enough writing, i.e. failure to highlight explicitly enough what indeed is the major breakthrough. But it might well be that the study is great, but not groundbreaking enough for this journal. I suggest the authors are given an opportunity to address this point.

More minor points:

I have doubts about the reasons for the traits considered. Some of them are not properly justified or are at risk of being

trivially correlated with each other, therefore potentially distorting the multivariate analyses. For example leaf mass and leaf area, or leaf C (which varies very little in general and, admittedly, see L 355, among climates in particular), especially considering they also look at leaf N and leaf dry mass content. And seed length and seed mass. Whereas rationale for each (or some) of the individual traits is summarily provided, there is no justification as to why this set of traits and not any other random list of traits were considered that are directly relevant to the specific questions in this study.

Why N and P are expressed per area and not as per mass, which is the expression used in the articles the authors use as basis, and in general more common in the trait literature? A hint is given around L. 260 but it should be made more explicit.

Why climatic variables explain more variance for woody than for non-woody plants (around L 270): this explanation is plausible, but the fact that woody plants have permanent organs that have to stand climate throughout the year for many years may also explain this tighter association.

Author Rebuttal letter:

The author's response to these comments can be found at the end of this file.

Version 1:

Reviewer comments:

Reviewer #1

(Remarks to the Author)

I greatly appreciate the effort that the authors have made with their careful revision. In the following, I will not comment on the points raised that I can agree with in the revised version and will only address open questions.

Fig. 1, L129: I suggest replacing "Colour bars indicate the value and direction of the axis 3" by "Gray scales indicate the loadings on axis 3" (see also Fig. S6). I would also delete "darker colours represent higher coordinate values. Correlations among traits, and between traits and climate variables, are represented by the direction of vectors" because "higher" is not meaningful with arbitrary axis orientation and I think the basic interpretation of PCA and RDA does not need to be explained.

L138-L142: "climatic influences on plants" is a little vague. The cited Prentice et al. 1992 refers in fact to a model to predict global patterns in vegetation physiognomy. It can be argued that the physiognomy of the vegetation is a good general proxy for most of the traits analyzed.

Fig. 3: The small font size remains challenging for the reader.

Fig. 4 and the weighted average calculation: I understand if the authors do not want to add and discuss these results, even though the differences clearly underline the caution with which the global trait maps should be used and interpreted.

L476-479: Please indicate which taxonomic backbone and corresponding nomenclature were used as a reference in this study. This would be particularly helpful for further use of the compiled data.

I find the new sentences on limitations important and helpful, but I would still be a little more cautious with the following two statements:

(1) L358: "These GAMs predicted up to 77% of global trait variation correctly" and L431 "[...] we can explain up to three-quarters (on average about a half) of global variation for all 16 FTs." As the data set does not represent all areas worldwide equally, the phrase "global trait variation" is somewhat misleading. A small addition like "variation in our global data sample" seems more appropriate.

(2) L382-390: "H was overestimated globally by the GAM compared to iNaturalist, especially in the northern high latitudes. Given that GAMs of these FTs showed good predictive performance ($R_{adj2} \geq 49\%$, see Table S11), one potential explanation lies in the probable sampling bias of iNaturalist. ...". sPlotOpen is also biased! A more balanced statement seems appropriate, because the good GAM performance is not a good reason to trust the GAM prediction in under-sampled regions more than iNaturalist, if the latter has many observations there (see Fig. 1 in Wolf et al. 2022).

Additional comments:

L18: For the phrase "values of LES traits were higher in drier climates" it seems better to mention the traits directly, because it makes a difference whether the reader thinks of LMA or SLA.

L94: I assume these traits are called 'major FTs' because they belong to the global spectrum of plant form and function? If so, this can be added.

L169: It could be added that the relative importance of the variables here refers to GAMs.

L231: It should be clear from the caption what "with above-average R²" refers to.

L287: I find this new discussion about SSD and the three life forms very interesting!

L523: It unclear to me how matching the trait dataset with the bioclimate affects the number of "natural vegetation plots"? How was the sPlotOpen variable "Naturalness" considered (indicates natural and semi-natural vegetation, but has also many NAs)? Further information on the vegetation types analyzed would be helpful. Different vegetation types could also have contributed to the differences detected with iNaturalist.

Author Rebuttal letter:

Reviewers' comments:

Reviewer #1 (Remarks to the Author):

I greatly appreciate the effort that the authors have made with their careful revision. In the following, I will not comment on the points raised that I can agree with in the revised version and will only address open questions.

Responses:

We are grateful for the time you have taken to review our manuscript and for your helpful comments. We have revised our manuscript according to your suggestions. Please check the responses to line-by-line comments below for more details. The latest changes in the revised manuscript are highlighted in red with prior modifications marked in blue.

Fig. 1, L129: I suggest replacing "Colour bars indicate the value and direction of the axis 3" by "Gray scales indicate the loadings on axis 3" (see also Fig. S6). I would also delete "darker colours represent higher coordinate values. Correlations among traits, and between traits and climate variables, are represented by the direction of vectors" because "higher" is not meaningful with arbitrary axis orientation and I think the basic interpretation of PCA and RDA does not need to be explained.

Responses:

We have revised the captions of Fig. 1 (MS L130 – L132), Fig. S6 (SI L69), Fig. S16 (SI L167 – L168) and Fig. S17 (SI L195 – L196) according to your recommendations:

"Gray scales indicate the loadings on the third axis. Solid arrows (with blue labels in RDA) represent traits and dashed arrows (with red labels in RDA) represent bioclimatic variables."

L138-L142: "climatic influences on plants" is a little vague. The cited Prentice et al. 1992 refers in

fact to a model to predict global patterns in vegetation physiognomy. It can be argued that the physiognomy of the vegetation is a good general proxy for most of the traits analyzed.
Responses:

We have modified the statement (L137 – L140) according to your suggestions. Since we have explained the reasons for choosing these variables in the Methods (L498 – L503), we would not go into details here.

“We calculated three bioclimatic variables for each plot (Fig. S2), which represent the three major independent climatic controls on the global geographic distribution of vegetation physiognomy^{50–52} and therefore likely also the global distribution of key plant traits⁵² (see Methods). The three bioclimatic variables are:...”

Fig. 3: The small font size remains challenging for the reader.

Responses:

Thanks for the suggestion. Please check the updated Fig. 3 (L196).

Fig. 4 and the weighted average calculation: I understand if the authors do not want to add and discuss these results, even though the differences clearly underline the caution with which the global trait maps should be used and interpreted.

Responses:

Thank you for your suggestion. We have now explained this in detail in the Methods (L584 – L595).

“Although using a weighted average of three maps from different plant groups should be a more straightforward way to generate global maps, we did not adopt this approach. This decision was made for the following reasons: (1) Simulating separate global trait patterns for each of the three plant groups, and calculating weighted averages of them with fractional cover of plant groups as weights, would only be applicable where traits possess species-level values. In this study, only the six major traits are provided with species-level mean values. The remaining traits have plot-level mean values, which are weighted averages of trait values for all plant taxa within the plot. It is almost impossible to predict global trait patterns for each plant group from such community weighted mean trait values.

However, using the GAMs including fractional covers of plant groups provides a more universal alternative approach, which is directly applicable to plot-level means. (2) A simple weighted average of three maps in three plant groups, using fractional cover of plant groups as weights, would fail to account for the unavoidable interaction between bioclimates and vegetation cover.”

L476-479: Please indicate which taxonomic backbone and corresponding nomenclature were used as a reference in this study. This would be particularly helpful for further use of the compiled data.

Responses:

We now have clarified this in Methods (L486 – L487):

“The scientific names of all taxa from different datasets were aligned before integration by using Taxonomic Name Resolution Service V5.2 (TNRS, <https://tnrs.biendata.org/>)⁸¹.”

I find the new sentences on limitations important and helpful, but I would still be a little more cautious with the following two statements:

(1) L358: "These GAMs predicted up to 77% of global trait variation correctly" and L431 "[...] we can explain up to three-quarters (on average about a half) of global variation for all 16 FTs." As the data set does not represent all areas worldwide equally, the phrase "global trait variation" is somewhat misleading. A small addition like "variation in our global data sample" seems more appropriate.

Responses:

We have changed the relevant statements according to your suggestion:

L364 – L365:

“These GAMs predicted up to 77% of variation in our global data sample correctly.”

L436 – L438:

“Based on three bioclimatic variables and global vegetation cover and their interaction effects, we can explain up to three-quarters (on average about a half) of global variation of community-weighted means for all 16 FTs.”

(2) L382-390: "H was overestimated globally by the GAM compared to iNaturalist, especially in the northern high latitudes. Given that GAMs of these FTs showed good predictive performance ($\text{Rad}j^2 \geq 49\%$, see Table S11), one potential explanation lies in the probable sampling bias of iNaturalist. ...". sPlotOpen is also biased! A more balanced statement seems appropriate, because the good GAM performance is not a good reason to trust the GAM prediction in under-sampled regions more than iNaturalist, if the latter has many observations there (see Fig. 1 in Wolf et al. 2022).

Responses:

Thank you for your suggestion. We have added a more balanced statement in L390 – L394:

“Plot-level mean values of 16 FTs in our study were obtained from the sPlotOpen dataset. In

principle, the above disagreements may arise from the respective sampling biases of iNaturalist and sPlotOpen. The iNaturalist contributors are by no means evenly sampled across growth forms (tree/shrub/herb)⁴¹, while the sPlotOpen data cover a relatively small portion of the global climate space.”

Additional comments:

L18: For the phrase "values of LES traits were higher in drier climates" it seems better to mention the traits directly, because it makes a difference whether the reader thinks of LMA or SLA.

Responses:

Here are the revised descriptions (L18 – L19):

“Size traits (plant height, diaspore mass) however were generally higher in warmer climates, while LES traits (leaf mass and nitrogen per area) were higher in drier climates.”

L94: I assume these traits are called 'major FTs' because they belong to the global spectrum of plant form and function? If so, this can be added.

Responses:

Yes, and here is the revised statement (L94 – L95):

“We refer to these six as ‘major FTs’ as they also form the global spectrum of plant form and function.”

L169: It could be added that the relative importance of the variables here refers to GAMs.

Responses:

We have added the detail in the caption of Fig. 2 (L170 – L172):

“Fig. 2 | Relative importance of each bioclimatic variable in predicting six major plant functional traits of non-woody (a), woody deciduous (b) and woody evergreen (c) plants derived from Generalized Additive Models (GAMs).”

L231: It should be clear from the caption what "with above-average R²" refers to.

Responses:

We have revised the caption of Fig. 4 (L236 – L240) and added more details in Results (L215 –

L217):

L236 – L240:

“Global patterns of nine traits produced by the best-fit generalized additive models ($R^2 \geq 49\%$) based on three bioclimatic variables combined with remotely-sensed fractional cover data for the three plant groups (see Methods for details). The best-fit model here refers to the model with adjusted R^2 above the median level of the adjusted R^2 over all 16 FTs.”

L215 – L217:

“These GAMs explained up to 77% of the global variation of plot-level mean trait values, with a median adjusted R^2 of 52% (average median calculated from 54% and 49%) over all 16 FTs.”

L287: I find this new discussion about SSD and the three life forms very interesting!

Responses:

Thank you for this positive feedback!

L523: It unclear to me how matching the trait dataset with the bioclimate affects the number of "natural vegetation plots"? How was the sPlotOpen variable "Naturalness" considered (indicates natural and semi-natural vegetation, but has also many NAs)? Further information on the vegetation types analyzed would be helpful. Different vegetation types could also have contributed to the differences detected with iNaturalist.

Responses:

We now have clarified this in Methods (L528 – L533):

“We checked the ‘naturalness’ of each vegetation plot in our trait dataset against the global map of land cover provided by the ESA CCI-LC database (<https://www.esa-landcover-cci.org/?q=node/164>) at 300 m spatial resolution in 2010. For plots belonging to all types of croplands, urban areas, bare areas, water bodies and permanent snow and ice in the ESA CCI-LC database, we defined them as ‘unnatural vegetation’ and removed those plots from the trait dataset used for data analysis.”

Additional comment from the editor to the Author:

Please note that your response to reviewer #2 was assessed by reviewer #1. Your response to the comments made by reviewer #3 was assessed by our team of in-house editors.

Responses:

We sincerely appreciate your professional review work on this manuscript!

Version 2:

Reviewer comments:

Reviewer #1

(Remarks to the Author)

Thank you for this revised version which has improved considerably. My earlier comments have been taken into account. Only a few comments remain about the content - and other are only related to an improved readability of text and figures.

L121-122: I would change "SSD in non-woody plants was strongly correlated with LMA and Narea." to "... correlated with LA and Narea." change. According to Fig. 1d, LA should be more strongly (negatively) correlated with SSD than LMA - and even more strongly than with Narea.

L162-L168: I was a bit confused when I compared this paragraph with Fig. 2. For example, why is MGST highlighted for both woody species groups, but not for the non-woody? The rationale for the examples chosen might be not clear to the reader and appears somewhat arbitrary.

L185-187: I struggled again with where to find MTCO < 4°C in Figures 3b, 3e and 3h. Despite the earlier explanation, I would stick to the values shown, e.g. "(MTCO = 2°C and lower in Fig. 3b)".

L262, caption of Fig. 5: The red line visualizes a linear regression and should not be labeled correlation.

L269, caption of Fig. 5: "showed higher agreement" - compared to what? Should be clear from the caption.

L297-299: This remaining sentence from the previous version now does not seem to be well connected to the new paragraph and could be deleted.

L299- 300: "SSD was considered to be independent of other FTs" is misleading because as stated by Diaz et al. "Stem specific density (SSD) and leaf area (LA) also load heavily on the plane and are correlated with both the H-SM and the LMA-Nmass dimensions". Therefore, although SSD is not stronger correlated with a single of the two major trait dimensions it is not independent of the other FTs.

L332-333: "The short lifespan of non-woody plants ...". Here, I suggest adding "of many non-woody plants" because some non-woody plants can become older than other woody plants and there is not a strict separation by age.

L386-389: It is difficult to compare the mentioned findings in Fig. S14, as the color gradients of the maps do not consistently show over- and underestimations (zero does not consistently separate red from blue colors). Please use a consistent color gradient for a consistent visualization of the over- and underestimated areas.

L399-410: Please correct "sPlotOpen is an aggregation of trait data from various studies ..." by "sPlotOpen is an aggregation of vegetation plots from various studies ...". If I understand correctly, the trait data of sPlotOpen are not measured for the species on the vegetation plots and are TRY-based instead. The sentence "not all species within every vegetation plot have had their trait values measured" might easily be misunderstood by some readers. They trait data share therefore the same bias in species coverage and the bias introduced by the gap-filling algorithm used by TRY as iNaturalist or the Diaz et al. data.

L417-418: Could be formulated more generally, like "However, some FTs can be influenced by other factors in addition to the bioclimate used." This also includes more trait-specific climatic factors that are excluded in the current version.

L458: Please specify where this information can be found within the supplementary.

L468-469, L484, L613: The term "growth form" should be a better fit for "plant form". Despite the title, Diaz et al. 2022 used the term "growth form" too.

L560-561, about concurvity in GAMs: According to the newly added information, MMGST = 0.867 is high and the variable importance for the individual GAMs must be interpreted with caution. This statistical limitation might be added to the discussion.

L593-595, about why not using weighted averaging: I completely agree with (1), but I don't really understand (2). Since (1) is convincing, (2) might be deleted?

L601-602: "we spatially rescaled our global trait maps into 2° resolution" - please specify how the data were aggregated.

Table S12: Please change the percentage values to the same format of the other values.

Author Rebuttal letter:

Reviewers' comments:

Reviewer #1 (Remarks to the Author):

Thank you for this revised version which has improved considerably. My earlier comments have been taken into account. Only a few comments remain about the content - and other are only related to an improved readability of text and figures.

Responses:

We deeply appreciate the careful review of the manuscript and the helpful suggestions you provided. We have revised our manuscript according to your suggestions. The latest changes in the revised manuscript are highlighted in purple with prior modifications marked in red (second revisions) and blue (first revisions).

L121-122: I would change "SSD in non-woody plants was strongly correlated with LMA and Narea." to "... correlated with LA and Narea." change. According to Fig. 1d, LA should be more strongly (negatively) correlated with SSD than LMA - and even more strongly than with Narea.

Responses:

We apologize for any confusion caused by not clearly indicating in the text that the results presented here are derived from PCA rather than RDA. We have now added the labels of figure panels in the manuscript (L121 – L124).

"SSD in non-woody plants was strongly correlated with LMA and Narea (Fig. 1a). SSD in woody evergreen plants was more correlated with plant size (Fig. 1c), while SSD in woody deciduous plants showed an intermediate pattern (Fig. 1b)."

L162-L168: I was a bit confused when I compared this paragraph with Fig. 2. For example, why is MGST highlighted for both woody species groups, but not for the non-woody? The rationale for the examples chosen might be not clear to the reader and appears somewhat arbitrary.

Responses:

We realized that this text was hard to follow and have rewritten it in what we hope is a clearer way (L163 – L171). We also removed the caveat about distinguishing winter temperature effects, as we now think this point is unnecessary.

"Across all plant groups H variation was dominated by growing-season temperature, and LMA and Narea variation by moisture availability (with an additional influence of growing-season temperature

on LMA in non-woody plants). DM variation was dominated by moisture availability in non-woody plants but by growing-season temperature (with an additional influence of winter temperature) in woody deciduous plants. By contrast, variation in SSD and LA showed similar controls in non-woody and woody deciduous plants (dominated by moisture availability and growing-season temperature for SSD and moisture availability for LA), but a different type of control – dominated by growing-season and winter temperatures – for woody evergreen plants.”

L185-187: I struggled again with where to find $MTCO < 4^{\circ}\text{C}$ in Figures 3b, 3e and 3h. Despite the earlier explanation, I would stick to the values shown, e.g. "($MTCO = 2^{\circ}\text{C}$ and lower in Fig. 3b)".

Responses:

Thanks for the suggestion. We have modified the statement (L185 – L192):

“For example, among FTs related to the LES, Fig. 3b, e and h shows that LMA in all plant groups increases towards drier climates – but LMA in non-woody plants tends to decrease with growing-season temperature in climates with cold winters ($MTCO \leq -2^{\circ}\text{C}$ in Fig. 3b). LMA in woody plants is less sensitive to temperature (Fig. 3e and h); however woody deciduous plants show highest LMA in climates with warmer winters ($MTCO \geq 10^{\circ}\text{C}$ in Fig. 3e), while woody evergreen plants show highest LMA in climates with colder winters ($MTCO \leq -2^{\circ}\text{C}$ in Fig. 3h). Among FTs representing plant size (Fig. 3c, f and i), H is greatest in climates that are both wet ($\ln MI > 1$) and warm ($MGST > 15^{\circ}\text{C}$, $MTCO \geq 10^{\circ}\text{C}$) within all groups.”

L262, caption of Fig. 5: The red line visualizes a linear regression and should not be labelled correlation.

Responses:

We have modified the captions of Fig. 5 (L263 – L276) and Fig. S13 (L138 – L145) according to your suggestions:

MS L265 – L266 and SI L140 – L142:

“Red lines visualise the linear regression between model-predicted map pixel values and the iNaturalist map pixel values estimated at a 2° spatial resolution.”

MS L275:

“...parameters for evaluating linear regressions for all 16 traits...”

L269, caption of Fig. 5: "showed higher agreement" - compared to what? Should be clear from the caption.

Responses:

We have revised the caption of Fig. 5 (L271 – L273):

“The nine traits displayed here showed higher agreement between model-predicted maps and the iNaturalist maps compared to the remaining traits ($R^2 \geq 13\%$ and slope > 0.5 and < 2).”

L297-299: This remaining sentence from the previous version now does not seem to be well connected to the new paragraph and could be deleted.

Responses:

Thank you for this comment. This sentence is not actually out of context, but we have made a small modification which explains how it is linked to SSD (L299 – L302).

“This novel finding is explicable in adaptive terms. In dry climates, some herbaceous and succulent plants (typically with low SSD) can fix CO₂ through their green stems, reducing water loss from the leaves and providing an extra source of carbon^{61–63}. ”

L299- 300: "SSD was considered to be independent of other FTs" is misleading because as stated by Diaz et al. "Stem specific density (SSD) and leaf area (LA) also load heavily on the plane and are correlated with both the H–SM and the LMA–Nmass dimensions". Therefore, although SSD is not stronger correlated with a single of the two major trait dimensions it is not independent of the other FTs.

Responses:

Thank you for your suggestion. We have modified the statement (L302 – L303):

“The variation of SSD did not show stronger correlation with a single of the two major trait dimensions in the global spectrum of plant form and function¹².”

L332-333: "The short lifespan of non-woody plants ...". Here, I suggest adding "of many non-woody plants" because some non-woody plants can become older than other woody plants and there is not a strict separation by age.

Responses:

We have revised the statement according to your suggestion (L335 – L337):

“The short lifespan of many non-woody plants also represents an adaptive strategy that facilitates their survival in demanding environments.”

L386-389: It is difficult to compare the mentioned findings in Fig. S14, as the color gradients of the maps do not consistently show over- and underestimations (zero does not consistently separate red from blue colors). Please use a consistent color gradient for a consistent visualization of the over- and underestimated areas.

Responses:

We have changed Fig. S14 (L146 – L149) and Fig. S23 (L245 – L248) according to your suggestions:

L399-410: Please correct "sPlotOpen is an aggregation of trait data from various studies ..." by "sPlotOpen is an aggregation of vegetation plots from various studies ...". If I understand correctly, the trait data of sPlotOpen are not measured for the species on the vegetation plots and are TRY-based instead. The sentence "not all species within every vegetation plot have had their trait values measured" might easily be misunderstood by some readers. They trait data share therefore the same bias in species coverage and the bias introduced by the gap-filling algorithm used by TRY as iNaturalist or the Diaz et al. data.

Responses:

Thank you for your suggestion. We have revised the relevant statements.

L402 – L404:

"sPlotOpen is an aggregation of vegetation plots from various studies with specific and distinct research aims across global ecosystems that do not share the same criteria in the sampling approach, which may also lead to biases in this dataset^{37,38}."

L407 – L408:

"Furthermore, while sPlotOpen provides plot-level means for each trait, not all species within every vegetation plot have been recorded."

L417-418: Could be formulated more generally, like "However, some FTs can be influenced by other factors in addition to the bioclimate used." This also includes more trait-specific climatic factors that are excluded in the current version.

Responses:

We have revised the sentence according to your suggestion (L420 – L421):

“However, some FTs can be influenced by other factors in addition to the bioclimatic variables used in this study.”

L458: Please specify where this information can be found within the supplementary.

Responses:

We have clarified the statement (L460 – L461):

“...(see Supplementary Information SI 2 for details).”

L468-469, L484, L613: The term "growth form" should be a better fit for "plant form". Despite the title, Diaz et al. 2022 used the term "growth form" too.

Responses:

We have replaced “plant form” with “growth form” in the updated Manuscript and Supplementary Information.

L560-561, about concavity in GAMs: According to the newly added information, MGST = 0.867 is high and the variable importance for the individual GAMs must be interpreted with caution. This statistical limitation might be added to the discussion.

Responses:

Thank you for the helpful comment. We have added the statistical limitation in Methods (L564 – L570):

“We then calculated the relative importance of each bioclimatic variable in the GAMs to determine which of the predictors is more significantly related to the distribution of each FT. The concavity of three bioclimatic variables in GAMs was checked before fitting models (mean values: In MI = 0.487 ± 0.06 , MTCO = 0.651 ± 0.07 , MGST = 0.867 ± 0.02). MGST presented relatively high concavity, which make it challenging to interpret the individual importance of variables in trait prediction. Nonetheless, the overall predictive performance of the GAMs remains robust, with the combined effect of the predictors effectively capturing the variation in FTs.”

L593-595, about why not using weighted averaging: I completely agree with (1), but I don't really understand (2). Since (1) is convincing, (2) might be deleted?

Responses:

Thank you for your suggestion. We have deleted the (2) (L591 – L600):

“Although using a weighted average of three maps from different plant groups should be a more straightforward way to generate global maps, we did not adopt this approach. This is because simulating separate global trait patterns for each of the three plant groups, and calculating weighted averages of them with fractional cover of plant groups as weights, would only be applicable where traits possess species-level values. In this study, only the six major traits are provided with species-level mean values. The remaining traits have plot-level mean values, which are weighted averages of trait values for all plant taxa within the plot. It is almost impossible to predict global trait patterns for each plant group from such community weighted mean trait values. However, using the GAMs including fractional covers of plant groups provides a more universal alternative approach, which is directly applicable to plot-level means.”

L601-602: "we spatially rescaled our global trait maps into 2° resolution" - please specify how the data were aggregated.

Responses:

We have clarified the aggregation method (L606 – L610):

“Therefore, we aggregated and reprojected our global trait maps according to the iNaturalist maps (2° spatial resolution) using bilinear interpolation by ‘projectRaster()’ function in the R raster package93. Then we assessed the pixel-by-pixel agreement between our trait maps with the iNaturalist trait maps by linear regression using the ‘lm()’ function in the R stats package.”

Table S12: Please change the percentage values to the same format of the other values.

Responses:

We have changed the 28% into 0.2800 in the Table S12.

Version 3:

Reviewer comments:

Reviewer #1

(Remarks to the Author)

The comments from the last round have been carefully and comprehensively considered by the authors. I agree with the changes and hope that the authors found my comments constructive and helpful.

One last minor comment: While reading through, it took me again some time to follow the relationships described in Figure 3. This particularly applied to the half-sentence "LA was less sensitive to moisture in woody plants (Fig. 3d and g) compared to non-woody plants (Fig. 3a)" (L193-L195). Although confirmed by Figure 1d-f, I couldn't clearly follow this in Figure 3. The half-sentence also seems to contradict the "strong relationships" mentioned before and it might therefore be deleted or referred to Fig. 1d-f.

Author Rebuttal letter:

Reviewers' comments:

Reviewer #1 (Remarks to the Author):

The comments from the last round have been carefully and comprehensively considered by the authors. I agree with the changes and hope that the authors found my comments constructive and helpful.

One last minor comment: While reading through, it took me again some time to follow the relationships described in Figure 3. This particularly applied to the half-sentence "LA was less sensitive to moisture in woody plants (Fig. 3d and g) compared to non-woody plants (Fig. 3a)" (L193-L195). Although confirmed by Figure 1d-f, I couldn't clearly follow this in Figure 3. The half-sentence also seems to contradict the "strong relationships" mentioned before and it might therefore be deleted or referred to Fig. 1d-f.

Responses:

We appreciate the time and effort you have dedicated to reviewing our manuscript. Your insightful comments and constructive feedback have been invaluable in enhancing the quality of our work. The latest changes in the revised manuscript are highlighted in orange, with prior modifications marked in purple (third revisions), red (second revisions) and blue (first revisions).

We have revised the statement according to your suggestion in L193 – L195:

"LA showed a general pattern of increase with moisture within all three plant groups (Fig. 3a, d and g), but LA was less sensitive to moisture in woody plants (Fig. 3d and g) compared to non-woody plants (Fig. 3a; also confirmed by Fig. 1d–f)."

Responses to comments:

Reviewer #1:

BRIEF SUMMARY

The present work aims to generate and test global maps for 16 important plant functional traits. For this purpose, generally available global data from sPlotOpen (vegetation plots with species abundances and community weighted traits), from the "global spectrum of plant form and function" (compiled traits at species level), and iNaturalist (species occurrences and compiled trait maps) are used along with global climate (CHELSA, Worldclim) and land cover data (ESA CCI-LC). For six major traits maps are also modeled separately for the three groups of non-woody, woody deciduous and woody evergreen plants. The maps are finally compared with independent traits values based on iNaturalist (and TRY), confirming the general spatial trait patterns.

OVERALL IMPRESSION

Although global maps of individual plant traits already exist, the present work is comprehensive and additionally distinguishes three different growth forms. On the other hand, and in addition to the more specific comments below, I missed (1) a clear overview of the workflow including the directly used vs. recalculated traits, major traits vs. other traits, trait maps for plant groups vs. all plants, (2) a more detailed justification of the GAM variable selection for the global predictions, and (3) a discussion of the limitations of this work. Overall, however, I believe that a revised version can make a stimulating and valuable contribution to the current discussion on global patterns of plant functional traits, their drivers, and possible future dynamics.

Responses:

We appreciate this very positive review. We have added more details regarding the three aspects of concern. To address point (1), we have created a flow chart that provides an overview of this study (Fig. S1). For point (2), we have explained the reasons for variable selection for the GAMs in Methods (L490 – L495). For point (3), we have clarified the limitations of this work in the Discussion (L391 – L424). More details are provided below.

SPECIFIC COMMENTS

- page 3: I missed the information about previous approaches to global maps of similar traits (see comment in page 17 below).

Responses:

We have added more details about previous approaches to global trait mapping in Main (L63 – L71):

“Although satellite remote sensing offers potential to directly map plant traits at large scales^{28–30}, this approach can only retrieve a limited set of traits from leaves and upper canopies, with moderate accuracy²⁸. In addition, statistical upscaling and machine learning approaches based on relationships between FTs and environmental factors have recently been used to produce global maps for a few (mainly leaf) FTs^{2,3,25,29,31–34}. A recently published work simulated global maps of three leaf traits via optimality models based on eco-evolutionary optimality theory³⁵. However, the various published global trait maps do not always show consistent global patterns, reflecting differences in data sources and upscaling methods^{35,36}.”

- page 4: The number of species is impressive, but not all traits are available for all species. An extended Table 1 in the Appendix would be helpful - with information on the number of species per trait with available trait values and the origin of the pixel values used as response for the GAM calibration and evaluation (directly from sPlotOpen or iNaturalist publications vs. calculated for this study).

Responses:

We have added Table S1, showing the summary of sample sizes, in Supplementary Information:

Table S1 | Sample size of all data used in this study. ^a Plant functional traits used in this study. ^b Abbreviation and ^c unit of each of the 16 traits. ^d Number of plots having plot-level mean trait values. ^e Number of species having species-level mean trait values. ^f Number of pixels of iNaturalist trait maps at 2° spatial resolution. ^{g–h} Number of pixels of global trait maps upscaled from generalised

additive models (GAMs) at ^g 0.1° spatial resolution and ^h 2° spatial resolution. ⁱ Number of pixels having both iNaturalist trait values and GAM trait values when making comparisons at 2° spatial resolution. The first six traits are major traits having both species-level¹ and plot-level trait² means. The remaining traits only have plot-level trait means (community-weighted means, CWMs)².

Trait ^a	Abbreviation ^b	Unit ^c	N _{plot} ^d	N _{species} ^e	N _{iNaturalist} ^f	N _{GAM01} ^g	N _{GAM2} ^h	N _{Comparison} ⁱ
Leaf area	LA	mm ²		35,773	3,381			1,851
Leaf mass per unit area	LMA	kg/m ²			3,573			1,916
Leaf nitrogen content per unit area	N _{area}	g/m ²			3,433			1,871
Stem specific density	SSD	g/cm ³			3,198			1,802
Plant height	H	m			3,732			1,980
Diaspore mass	DM	mg			3,749			1,987
Leaf fresh mass	LFM	g			2,476			1,417
Leaf phosphorus content per unit area	P _{area}	g/m ²			3,421			1,856
Leaf carbon content	C _{mass}	mg/g	77,074		3,326	830,086	2,397	1,818
Leaf dry matter content	LDMC	g/g			3,286			1,800
Stem conduit (vessel and tracheid) element length	WVL	µm			2,314			1,320
Stem conduit density	SCD	mm ⁻²		0	2,869			1,622
Seed number per reproductive unit	SN	NA			2,707			1,502
Seed length	SL	mm			2,745			1,528
Dispersal unit length	DUL	mm			2,818			1,534
Leaf nitrogen isotope ratio	δ ¹⁵ N	per million			2,774			1,602

- page 5, Table 1 - just as a remark: Although sufficiently justified in the manuscript, it is still a pity that N_{area} and not N_{mass} was used. This makes a direct comparison with Diaz et al. 2016 more difficult.

Responses:

We have now explained the reasons for not focusing on N_{mass} and P_{mass} in Methods (L445 – L452):

“Here we focus on N_{area} and P_{area}, because Osnas et al.⁷⁹ showed that leaf nitrogen and phosphorus contents are approximately distributed proportional to leaf area rather than mass. N_{area}

and P_{area} were found to be proportional to LMA and photosynthetic capacity of plants^{59,60,80}. Both LMA and photosynthetic capacity are quantitatively predictable from climate^{45,80}, which may lead to potentially high predictability of N_{area} and P_{area} from climates. In contrast, values of N_{mass} and P_{mass} may be more conservative under climate change. We also conducted the following analyses on N_{mass} and P_{mass} (see Supplementary Information for details) and found that climate is not good predictor of their distributions.”

We have also provided the results of analyses on N_{mass} and P_{mass} in Supplementary Information. Please check *SI 2 Analysis for mass-based leaf nitrogen content (N_{mass}) and leaf phosphorus content (P_{mass})* for more details.

- page 7, Fig. 1: Because the orientation of PCA and RDA axes is arbitrary, please check to flip the axes of each plot to show in all plots the same axes orientation. For example, the two axes of plot b and c point in opposite directions. To facilitate the comparison, Fig. 2 of Diaz et al. 2016 might be used as reference.

Responses:

Here is the updated Fig. 1: The orientation of PCA and RDA axes has been rotated according to Fig. 2 of Diaz et al. 2016.

Fig. 1 | Principal component analysis (PCA) (a–c) and redundancy analysis (RDA) (d–f) for six major plant functional traits of non-woody (a, d), woody deciduous (b, e) and woody evergreen (c, f) plants. Six major traits are leaf area (LA, mm²), leaf mass per unit area (LMA, kg/m²), leaf nitrogen content per unit area (N_{area}, g/m²), stem specific density (SSD, g/cm³), plant height (H, m) and diaspore mass (DM, mg). The orientation of axes has been rotated according to the Figure 2 in ref. 12. Colour bars indicate the value and direction of the axis 3 for PCA and RDA; darker colours represent higher coordinate values. Correlations among traits, and between traits and climate variables, are represented by the direction of vectors: solid arrows (with blue labels in RDA) for traits and dashed arrows (with red labels in RDA) for bioclimatic variables. All six traits were natural-log transformed before analysis. Both log-transformed traits and bioclimatic variables were rescaled to a mean of 0 and a standard deviation of 1 before analysis. In MI, log-transformed moisture index; MTCO, mean temperature of the coldest month; MGST, mean growing-season temperature (see Methods for definition).

- page 7: The use of only three bioclimatic variables and the same variables for all FTs can be questioned and requires some justification. Without question, the three bioclimatic variables have important impacts on plants. However, this does not necessarily apply to individual traits, which most likely differ in their bioclimatic drivers.

Responses:

We have clarified the reasons for the selection of three bioclimatic variables in the updated manuscript:

Results (L138 – L142): “We calculated three bioclimatic variables for each plot (Fig. S2), representing the three major independent climatic influences on plants^{50–52}: mean growing-season temperature (MGST, °C, representing summer warmth), mean temperature of the coldest month (MTCO, °C, representing winter cold), and the log-transformed moisture index (ln MI, unitless, representing plant-available moisture: see Methods).”

Discussion (L406 – L411): “Our study applied a consistent set of three bioclimatic variables to each FT, thereby preserving model simplicity and providing a consistent baseline for comparison. The three selected variables reflect three most important aspects of climate that govern plant

distributions, i.e., winter cold, summer warmth and moisture availability⁵⁰⁻⁵². These three variables effectively explained global patterns of different FTs and sufficiently distinguished the main differences in trait-climate relationships among three plant groups.”

Methods (L490 – L495): “The three selected variables correspond to the three recognised climatic dimensions that regulate global geographic distribution of vegetation⁵⁰⁻⁵². They control the plant distribution limits by affecting the plant attributes that determine physiological processes and adaptive strategies, such as life form, leaf phenology, leaf size and stomatal conductance⁵⁰⁻⁵². Hence, the distribution of some key plant traits is likely to be a function of these bioclimatic variables, and thus may be quantitatively predicted by these variables⁵².”

- page 8, GAM calibration: Temperature related bioclimatic variables are often highly correlated. This seems also the case for MGST and MTCO as indicated by the RDA plots in Figure 1. Did you check for collinearity and how did you deal with it?

Responses:

It is true that MTCO and MGST are highly correlated with each other ($r = 0.81$). We assessed the “concurvity” (which is the non-linear form of collinearity) of three bioclimatic variables in GAMs before making predictions. The *mgcv* package in R provides a measure of concurvity and states in the documentation if such measures ≥ 1.0 then we should be concerned. Three indices can be used to evaluate concurvity. We choose the “estimate” index because it does not suffer from the pessimism or potential for over-optimism of the other two measures. All values of the index are less than 1 in GAMs with only three bioclimatic variables for all 16 traits: ln MI = 0.515, MTCO = 0.657, MGST = 0.867.

These results indicated that although MTCO and MGST are highly correlated, our GAM models are not affected by collinearity or concurvity issues. We are confident in the reliability and robustness of our analysis. And we now added the relevant statement in Methods (L550 – L551): “The concurvity of three bioclimatic variables in GAMs was checked before fitting models (ln MI = 0.515, MTCO = 0.657, MGST = 0.867) to ensure the reliability and robustness of the analysis.”

- page 9, Fig. 3: I was confused because "MTCO < 4°C" is not visible in Fig. 3 and it was also unclear what the MTCO values like "MTCO = -2 (°C)" mean - probably not an exact value, but a range of values? Fig. 3 is also very small to read. It might be better to focus on individual diagrams as examples or to move the entire figure to the Appendix.

Responses:

The response surfaces of trait values generated from GAMs were visualised as contours in the three-variable climate space in the form of two-dimensional slices. Each slice was created at an exact MTCO value (e.g. MTCO = -2 °C) rather than a range of values. We have added the explanation context in the caption of Fig. 3. Since there is such a pattern for MTCO = -2, -14 and -26 but not for MTCO = 10 and 22, we assumed that there might be a change in a value between -2 and 10, and the average value of 4 was taken.

We appreciate your suggestions on Fig. 3. However, because we aimed to compare different traits and different plant groups, putting several panels together better suited our purpose. Although we did not simplify the display of Fig. 3, we have changed the layout and caption of Fig. 3 to (hopefully) improve readability:

Fig. 3 | Examples of climate space diagrams of non-woody (a-c), woody deciduous (d-f) and woody evergreen (g-i) plants: showing distributions of natural-log transformed leaf area (LA, mm²) (a, d and g), leaf mass per unit area (LMA, kg/m²) (b, e and h) and plant height (H, m) (c, f and i) in the global climate space defined by three bioclimatic variables. Fitted trait values are presented as contours, with darker colours in the right colour bar representing higher trait values. Values of In MI and MGST vary continuously along the horizontal and vertical axes, respectively. Each slice is created at an exact MTCO value. Abbreviations and units of traits are shown in Table 1. In MI, log-transformed moisture index; MTCO, mean temperature of the coldest month; MGST, mean growing-season temperature (see Methods for definition). Climate space diagrams for all six major FTs are shown in Fig. S8.

- pages 11 and 23: The GAMs for the global maps included the fractional cover of two (not three) plant groups as additional predictors as well as their interaction terms with bioclimatic predictors. Although this seems reasonable, wouldn't a simple weighted average of the three maps of all three plant groups using fractional cover as weights be more straightforward? Did you tested such simple GAM ensembles (i.e., the weighted average of the three maps per trait in Fig. S8)?

Responses:

Indeed, using a weighted average of three maps from different plant groups should be a more straightforward way to generate global maps. Although we also tested this approach, we did not include the results in the manuscript. Instead, we fitted new GAMs that encompassed the fractional cover of plant groups to simulate global trait maps. This decision was made for the following reasons:

(1) Simulating separate global trait patterns for each of the three plant groups and calculating weighted averages of them with fractional cover of plant groups as weights to generate global trait maps are only applicable where traits possess species-level values. In this study, only six major traits have species-level mean values. The remaining 12 traits only have plot-level mean values, which are weighted averages of trait values for all plant taxa within the same plot. It is almost impossible to predict global trait patterns for each plant group from such community weighted mean trait values. Using the GAMs including fractional covers of plant groups should be a more universal approach that is applicable to different types of data (i.e. plot-level means).

(2) A simple weighted average of three maps in three plant groups using fractional cover of plant groups as weights fails to account for the interaction between bioclimates and vegetation cover. The interaction between bioclimatic variables and vegetation cover is inevitably present in reality and should be considered.

(3) Here are global trait maps generated by weighted average approach (we decided not to include them in the manuscript and appendix). Some outliers were observed in certain regions (such as LMA and N_{mass}), which could be attributed to the limited predictability of GAM for each trait in each plant group. This resulted in inaccurate trait value predictions in certain regions. Direct calculation of weighted averages may increase the inaccuracy of global trait simulation.

- page 12, Fig. 4: These maps are very small again. A 2x2 or 2x3 layout with only two columns seems more reader-friendly.

Responses:

Here is the revised Fig. 4:

Fig. 4 | Examples of global trait maps. Global patterns of nine traits with above-average R^2 ($\geq 49\%$) produced by the generalized additive models based on three bioclimatic variables combined with remotely-sensed fractional cover data for the three plant groups (see Methods for details). Colour gradients from blue to red indicate the increase in trait values. All traits are natural-log transformed. All maps are at 0.1° resolution. Global trait maps of all 16 functional traits are shown in Fig. S12. For maps in GeoTiff format, refer to the Data availability statement.

- page 17: "There are also previously published global trait maps generated from statistical modeling or machine learning methods". I missed this information in the introduction. In addition, remote sensing was used in previous studies. There is a recent publication with similar maps for leaf traits by the same last author: Dong et al. 2022, GEB, DOI: 10.1111/geb.13680. There, the overall global pattern for N.mass as an example is confirmed, but a comparison also shows some regional differences. A comparison with already published global trait maps would clearly improve the discussion section.

Responses:

Thank you for your helpful suggestions. We have added the relevant information in the Main and Discussion.

Main (L63 – L71): “Although satellite remote sensing offers potential to directly map plant traits at large scales^{28–30}, this approach can only retrieve a limited set of traits from leaves and upper

canopies, with moderate accuracy²⁸. In addition, statistical upscaling and machine learning approaches based on relationships between FTs and environmental factors have recently been used to produce global maps for a few (mainly leaf) FTs^{2,3,25,29,31–34}. A recently published work simulated global maps of three leaf traits via optimality models based on eco-evolutionary optimality theory³⁵. However, the various published global trait maps do not always show consistent global patterns, reflecting differences in data sources and upscaling methods^{35,36}.”

Discussion (L362 – L375): “Previously published global trait maps generated by statistical modelling or machine learning methods^{2,3,25,29,31–34} have only simulated global patterns for a subset of FTs we studied here; moreover, these maps show large differences in data sources, methodology and results^{35,36}. We also compared the general patterns of global trait maps generated herein with those from other published products for six major traits and LDMC. The predictability of our GAMs for the common six major traits was significantly stronger than models of other products (Table S12). Almost all six major traits presented similar global spatial patterns to that of previous studies. Nevertheless, regional differences emerged between our results and other products. For example, N_{area} values in Central Africa were at intermediate level in our GAM-predicted map (Fig. 4g), contrasting with the relatively lowest values observed in the theory-based map by Dong et al.³⁵; plant height (H) was predicted to be higher in Europe in this study compared to Schiller et al.’s predictions³³. The predictability of GAM of this study for LDMC was relatively weaker ($R_{\text{adj}}^2 = 0.39$, Table S12) and the global pattern of LDMC simulated in this study suggested significant deviation from the previous remote-sensing-based projections²⁹.”

- page 19: "We collected plant information including woodiness, plant growth form and leaf phenology for each taxon from regional floras, public databases and published papers." It remains unclear where the information mainly derived from. Because regional floras are mentioned first, it looks like the most important source (and a big effort!), but the Diaz et al. 2022 already includes woodiness, growth form, and leaf type.

Responses:

References of the plant information (including woodiness, plant growth form and leaf phenology) were listed in our plant form dataset (please check Data availability). We have added this

information in Methods (L455 – L458): “We collected plant information including woodiness, plant growth form and leaf phenology for each taxon from public databases, regional floras, and published papers (see Data availability for relevant references of plant information).”

We would like to clarify that the plant information used in this study was collected between June to August 2022, while the species-level trait dataset of Díaz et al. 2022 was published on 07 December 2022. We did not collect our plant form data directly from Díaz et al. 2022 but used the dataset to calibrate the plant information we collected. Additionally, the dataset of Díaz et al. 2022 does include woodiness, growth form and leaf type, but not leaf phenology (i.e. deciduous and evergreen).

- methods: A short comment on the taxonomic and nomenclature backbone of the various sources would be helpful. Were the raw data already harmonized?

Responses:

We have clarified this in Methods now (L476 – L479):

“After merging this species-level trait dataset with the sPlotOpen dataset and the plant form dataset, a total of 35,773 common species were explicitly classified into non-woody, woody deciduous and woody evergreen groups. The taxonomic backbone of different datasets was aligned before integration.”

- discussion of limitations: (1) Besides the partly very (too?) detailed discussion about the trait-trait and trait-climate relationships and about the iNaturalist evaluation, a discussion about limitations of the derived maps is missing. As stated in the sPlotOpen publication, these data "comes with a number of warnings" - first of all the sampling bias (as clearly shown in Fig. S2 of this MS). (2) Also the robustness of the global trait patterns remains unclear (e.g. when using a different variable selection for GAMs or a simple ensemble approach as mentioned above). (3) Already published global traits maps also show some differences worth to discuss (see comment on page 17 above). (4) And the plant groups of this MS are still very heterogeneous - e.g. in respect to life span (annuals vs. perennials) or evolutionary history (gymnosperms vs. angiosperms). On the other hand, the

comprehensive trait maps of this submission will provide a valuable data source for global studies in vegetation ecology and beyond.

Responses:

(1) We have followed these suggestions and added more details about limitations of data and models into the Discussion (L391 – L424).

(2) We have clarified the reasons for the selection of three bioclimatic variables in the updated Results (L138 – L142), Discussion (L406 – L411) and Methods (L490 – L495). Please check the responses to comments on page 7. We tested concurrency of three bioclimatic variables in GAMs before making predictions and have added the relevant statements in Methods (L550 – L551), which was listed in the responses to comments on page 8. We also explained the reason why we fitted new GAMs to generate global trait maps rather than using a more straightforward way (e.g. weighted average of three plant groups) in the responses to comments on pages 11 and 23.

(3) We conducted a general comparison between our global trait maps with previously published maps in the Discussion (L362 – L375). Please check the responses to comments on page 17.

(4) We also appreciate your comments on the classification of plant groups, but we decided not to add more classifications. Previous studies generally rely on multiple plant functional types for modelling. Our study aims to streamline plant functional grouping as much as possible to uncover primary distinctions, thereby substantially reducing model complexity. Evidence has demonstrated variations in functional traits among non-woody, woody deciduous and woody evergreen plants (as mentioned in Main L81 – L91). Consequently, we have categorized them into three primary functional groups.

Reviewer #2:

GENERAL COMMENTS

It was very interesting to read the article titled 'Global Patterns of Plant Functional Traits and Their Relationships to Climate.' Despite there being more studies modelling the spatial distribution of traits at a continental and global scale, as the authors pointed out, I find the authors' analysis across

different plant groups (e.g., woody and non-woody) highly valuable. However, there are several points throughout the paper that require substantial improvement.

Generally, several parts of the introduction lack the necessary ecological context for the concepts the authors mention (see detail recommendation in line by line comments). Furthermore, key methods and data deserve a concise yet improved explanation in the main text. For instance, it's challenging for readers to determine from where the climatic variables and remote sensing data used for analysis were obtained. Currently, readers might get the impression that the authors directly 'calculated' these variables, which is not the case. I also recommend improving the figures, their titles and captions to ensure that readers can readily grasp the key takeaways and main messages of the figures (see line by line comments).

The discussion could be streamlined. In some parts the authors should focus more on the meaning of the results, that is, avoid redundant findings or explanations of functional ecology concepts. Instead, contrast more your results with the ecological concepts and ideas.

Because 'global' studies can be controversial, I strongly suggest the authors to briefly address at the end the discussion limitations of the data used and its impact on the results. For instance, vast portions of the globe, such as the north of Africa (Fig. S2), lack any available plot. This affects the robustness of prediction of trait distribution for large areas.

It is amazing that the authors followed the key principle of the scientific method by sharing their code and data to make their study reproducible. But the authors need to revise the data related to the code because I could not find the files required to actually replicate the study (i.e., text, .xlsx., csv).

Lastly, I strongly advise minimizing the excessive use of abbreviations throughout the text, as it can make the document difficult to read. While I understand that certain abbreviations are essential, I advise to find a balance to prevent an overload of them. I also recommend avoid using passive voice when describing methods in the results section (see line by line comments).

Responses:

We have revised our manuscript according to these helpful recommendations, including more detailed description and explanation in the main text, informative figures and captions, detailed discussion of limitation of this work and minimizing excessive use of abbreviations and passive voice. Please check the responses to line-by-line comments below for more details. We hope the

quality and clarity of our work will be improved after revisions. We also added compiled data sets analysed in this study and [here is the updated link for data and code availability: https://zenodo.org/records/10462784.](https://zenodo.org/records/10462784)

LINE BY LINE COMMENTS

L 10-11: mention at least one strategy or adaptation that relate to ecosystems and their composition. Otherwise the starting of the abstract reads very vague.

Responses:

We have revised the starting of the Abstract as suggested (L10 – L11):

“Plant functional traits (FTs) determine growth, reproduction and survival strategies of plants adapted to their growth environment.”

L 18: what do the authors mean by traits being higher than others? Do you mean higher values, perhaps larger leaf areas or more acquisitive? A clarification is needed.

Responses:

We have clarified the statement (L18 – L20) as follows:

“Values of plant size traits were generally higher in warmer climates while values of LES traits were higher in drier climates, but larger leaves were associated principally with warmer winters in woody evergreens and with wetter climates in non-woody plants.”

L 27: Provide at least one trait that relates to the process you are mention afterwards, i.e., life history or responses to environmental change, to make the paragraph less vague.

Responses:

We have added examples of traits here (L27 – L31):

“Functional traits (FTs) – morphological, physiological and phenological characteristics of individual plants⁴ – determine plant growth, reproduction and survival strategies by influencing

life-history processes (e.g., seed traits), resource uptake and utilization (e.g., leaf nutrient contents), and responses to environmental change and disturbances (e.g., leaf traits and plant height)⁴⁻⁶.”

L 42: Explain what kind of function you are referring to, to avoid the text being vague.

Responses:

The explanation has been added (L43 – L48):

“At the community level, covarying FTs tend to perform interrelated functions^{2,16-18}. Wright et al.¹⁶ observed that wood density and leaf size are negatively correlated in neotropical forests. Lower wood density related with higher water conductivity per unit sapwood area and larger leaf area associated with potentially higher photosynthetic rate but also higher evapotranspiration. These two traits are thus interconnected through the hydraulic system governing water transport and its trade-off with photosynthetic efficiency¹⁶.”

L 43-44. Please specify what functions are meant here. It is not enough to say that quantifying is important. Also, it is not clear what 'their controls' mean.

Responses:

We have clarified the statement (L48 – L50) as follows:

“Quantifying the covariation of FTs and their controls is necessary for a full understanding of how the diversity of plants translates into community composition, productivity and adaptations to environment¹⁰.”

We did not clarify “their controls” here, as the environmental control was mentioned in the next paragraph.

L 48: briefly exemplify a type of shift.

Responses:

We have now exemplified the shifts (L52 – L55):

“...different combinations of FTs are selected for by environmental conditions^{5,14,20}, so climate change is expected to bring about shifts in species composition determined by their FTs, such as migration and extinction of needle-leaved trees in part of North America since the last glacial maximum²¹.”

L 100: Use active voice.

Responses:

Here is the revised statement (L111 – L112):

“We then used the CWMs to evaluate covariation among FTs within the non-woody, woody deciduous and woody evergreen groups.”

L 108-110: this is a very interesting result!

Responses:

Thank you for this positive feedback!

L 112: (Generally Figure 1) Figures should be able to stand alone. Include the abbreviation meanings in the figure caption, else readers won't understand the meaning of them. Remove the 's' in the word 'components'. Incorporate titles (non-woody, woody deciduous, etc) into the figure to make it easier to follow the results, see for example Figure 2 in Barajas Barbosa et al 2023 <https://www.nature.com/articles/s41586-023-06305-z/figures/2>

Responses:

We have now changed the display and caption of Fig. 1 as follows:

Fig. 1 | Principal component analysis (PCA) (a–c) and redundancy analysis (RDA) (d–f) for six major plant functional traits of non-woody (a, d), woody deciduous (b, e) and woody evergreen (c, f) plants. Six major traits are leaf area (LA, mm²), leaf mass per unit area (LMA, kg/m²), leaf nitrogen content per unit area (N_{area}, g/m²), stem specific density (SSD, g/cm³), plant height (H, m) and diaspore mass (DM, mg). The orientation of axes has been rotated according to the Figure 2 in ref. 12. Colour bars indicate the value and direction of the axis 3 for PCA and RDA; darker colours represent higher coordinate values. Correlations among traits, and between traits and climate variables, are represented by the direction of vectors: solid arrows (with blue labels in RDA) for traits and dashed arrows (with red labels in RDA) for bioclimatic variables. All six traits were natural-log transformed before analysis. Both log-transformed traits and bioclimatic variables were rescaled to a mean of 0 and a standard deviation of 1 before analysis. In MI, log-transformed moisture index; MTCO, mean temperature of the coldest month; MGST, mean growing-season temperature (see Methods for definition).

L 122: add the citation for this data so that readers can know the source of extracted the climatic variables.

Responses:

The three bioclimatic variables in this study were calculated from conventional climatic variables according to equations (6) – (8). Citations for the conventional climatic variables used to calculate the three bioclimatic variables have been added into the Methods (L500, L502, L504).

L 126: use active voice.

Responses:

Here is the revised statement (L142 – L143):

“We subsequently conducted a Redundancy Analysis (RDA) to describe the extent to which the six major FTs covary with these bioclimatic variables.”

L 131: instead of using the abbreviation, the text would be easier to read by mentioning the traits, meaning height and diaspore mass.

Responses:

Here is the revised statement (L147 – L151):

“Along the three axes, plant height and diaspore mass increased with the growing-season temperature (MGST) within all groups, and LMA and N_{area} decreased with increasing moisture (ln MI) within all groups. Plant size traits was positively correlated with winter cold (MTCO) on the first two axes, but they were negatively correlated on the third axis.”

L 137-139: it is unclear to which figure on the main text this analysis corresponds to. Use active voice here as well.

Responses:

Here is the revised statement (L156 – L158):

“We used Generalized Additive Models (GAMs) to quantify relationships between each major FT and bioclimatic variables in more detail, which allowed us to fit more complex surfaces and to distinguish growing-season from cold-month temperature effects (Figs 2–3).”

L 139-140: Complete the sentence and improve the explanation of why did you do this analysis. I guess you want to say that the relative importance of a climatic variable in relation to a trait, which is what you show in Figure 2.

Responses:

Here is the completed sentence (L158 – L161):

“We calculated relative importance values of explanatory variables (Fig. 2) in GAMs to quantify the individual contribution of three bioclimatic variables to the major FTs. These values measure the average percentage contribution of each variable in turn to the fit of the models based on all three variables.”

L 149: Again, do not use the abbreviation in the figure. You can use the whole name of the variable to make the figures more impactful. E.g., instead of using lnMI, write Moisture index. In the caption you can clarify that you log-transformed.

Responses:

Here is the revised Fig. 2:

Fig. 2 | Relative importance of each bioclimatic variable in predicting six major plant functional traits of non-woody (a), woody deciduous (b) and woody evergreen (c) plants. Six major traits are leaf area (LA, mm²), leaf mass per unit area (LMA, kg/m²), leaf nitrogen content per

unit area (N_{area} , g/m^2), stem specific density (SSD, g/cm^3), plant height (H, m) and diaspore mass (DM, mg). All six traits were natural-log transformed before the analysis. Moisture availability, log-transformed moisture index (ln MI, unitless); winter temperature, mean temperature of the coldest month (MTCO, $^{\circ}\text{C}$); growing-season temperature, mean growing-season temperature (MGST, $^{\circ}\text{C}$) (see Methods for definition).

L 154-155: please explain why are you doing this analysis and what is the key message of Figure 3?

Responses:

We have added the explanation (L178 – L179):

“This allowed for a more intuitive comparison of trait-climate relationships among the FTs and three plant groups.”

L 165: improve the readability of figure 3. Add distinctions between the panels and use them here to guide the reader to where exactly this result is found on the figure.

L 172: (Generally Figure 3) It is hard to understand to which analysis the figure corresponds to, I assume from the GAM, but this should be crystal clear for readers. The figure is generally very hard to follow, as it has many panels. You can add distinctions for the figure, e.g., for trait LMA add d, e and f panels, and for trait H add g, h and i panels. Use these distinctions to link the figure in the text. Explain what is the meaning of the legend on the right side of the spaces.

Responses:

We have made changes as follows:

Fig. 3 | Examples of climate space diagrams of non-woody (a-c), woody deciduous (d-f) and woody evergreen (g-i) plants: showing distributions of natural-log transformed leaf area (LA, mm²) (a, d and g), leaf mass per unit area (LMA, kg/m²) (b, e and h) and plant height (H, m) (c, f and i) in the global climate space defined by three bioclimatic variables. Fitted trait values are presented as contours, with darker colours in the right colour bar representing higher trait values. Values of In MI and MGST vary continuously along the horizontal and vertical axes, respectively. Each slice is created at an exact MTCO value. Abbreviations and units of traits are shown in Table 1. In MI, log-transformed moisture index; MTCO, mean temperature of the coldest month; MGST, mean growing-season temperature (see Methods for definition). Climate space diagrams for all six major FTs are shown in Fig. S8.

L 178: Explain exactly what remote sensed data/product are you using and cite the reference of origin of the data.

Responses:

There appears to be a typographical error in the manuscript. We did not use remote sensed data for these separate trait maps. We apologize for any confusion it may have caused. Here is the revised paragraph (L203 – L206):

“Global maps for the six major traits

For each of the three plant groups, we upscaled the GAMs based on three bioclimatic variables to generate global distribution maps for the six major FTs at 0.1° spatial resolution, separately. The separate trait maps for each plant group are shown in Fig. S9.”

L 206: (Generally Figure 4) I find this figure is the one that best matches your title and the one that readers are expecting to see. But the figure title is not informative yet. Try to improve the title by saying, e.g., ‘Global distribution of nine plant traits. Colours indicate increase in x trait value...etc.’. Indicate the meaning of the legend on the right side of each map. Give more relevance to the traits in the maps by avoiding abbreviations on the figure, there is enough space for you to write the name each trait on top of each map. Only mention in the caption that traits were log transform. Try to improve the aesthetics, i.e., plot a 3x3 panel figure and slightly increase the map sizes.

Responses:

We also changed the global trait maps in SI according to these suggestions:

Fig. 4 | Examples of global trait maps. Global patterns of nine traits with above-average R^2 ($\geq 49\%$) produced by the generalized additive models based on three bioclimatic variables combined with remotely-sensed fractional cover data for the three plant groups (see Methods for details). Colour gradients from blue to red indicate the increase in trait values. All traits are natural-log transformed. All maps are at 0.1° resolution. Global trait maps of all 16 functional traits are shown in Fig. S12. For maps in GeoTiff format, refer to the Data availability statement.

L 213: Explain to the readers why is it a bench mark and how the comparison will improve the message of your study.

Responses:

We added the reason for using iNaturalist data as benchmark (L238 – L240):

“Independent global trait maps⁴¹ generated by linking plant observations from the iNaturalist citizen-science project to trait measurements from the TRY database were used as a benchmark for our GAM-based predictions, as the iNaturalist data included all 16 FTs considered in our study.”

L 252: What does it mean 'same trait dimension'?

Responses:

We have clarified the statement as follows (L275 – L277):

“FTs from the same dimensions of trait covariation tended to be influenced by climate in similar ways (Figs 1–3 and S8).”

L 264-265: This is a very short part where you nicely link your results and explain the meaning of them. But the rest of the paragraph predominantly outlines existing knowledge with little linking to the contribution of your results.

Responses:

We have expanded this paragraph and added more details (**L287 – L307**):

“Our analysis also revealed a differentiation in trait combinations among the three plant groups, specifically in the covariation of SSD with other FTs. SSD in non-woody plants was strongly correlated with LES traits (Fig. 1a). SSD in woody deciduous plants was relatively independent (Fig. 1b), while SSD in woody evergreen plants was more correlated with plant size traits (Fig. 1c). This novel finding is explicable in adaptive terms. In dry climates, some herbaceous and succulent plants can fix CO₂ through their green stems, reducing water loss from the leaves and providing an extra source of carbon^{61–63}. The variation of SSD was considered to be independent of other FTs in the global spectrum of plant form and function¹². Previous studies were either conducted on global vascular plants¹² or analysed trait data at the community level by using community-weighted means^{9,14}. These studies did not identify the functional differences among the non-woody, woody deciduous and woody evergreen plants, consequently failing to detect the distinctive trait combinations associated with SSD. We also extracted pixel values from the GAM-based global maps of the six major traits in each plant group and performed PCA and RDA on them to test these observed trait combinations were still present in the predicted trait values (Fig. S16). The distinct patterns of SSD among the three plant groups were not evident in the predicted trait values – particularly in non-woody plants, where the previously strong correlation between SSD and LES traits was now absent (Fig. S16a). Trait data used in original PCA and RDA were calculated from the database of Díaz et al.⁴⁰ (see Methods) in which observed records for SSD are available only for a few non-woody species; missing values of SSD were imputed via leaf dry matter content (LDMC)⁴⁰, which was found to be closely related to both LMA and N_{area}⁹: perhaps accounting for the strong correlation between (LDMC-derived) SSD and the LES traits.”

L 277-281: This is a nice explanation to the readers of your results!

Responses:

Thank you!

Reviewer #3:

I applaud the focus, trying to predict coordinated sets of FTs rather than isolated ones, and trying to get a much better handle than before on three major growth forms. The analytical methods are sound, the graphs informative and the datasets used the best available in terms of traits and plots. The use of iNaturalist information as validation is clever and fresh. And by focusing on co-variation of traits, and how they are affected by climate and growth form, this study in parts touches upon a key issue: traits are coordinated together in whole plants, and it is the combination what has to be viable.

So I don't have major doubts or criticisms about the general focus, methods or results. I do, however, wonder about the fundamental advancement of the field that this article represents, which would merit publication in *Comms Biol* and not a good specialized journal. At the moment, I cannot see that very clearly. The concluding statements that this study shows that "it is useful to differentiate between non-woody, woody deciduous and woody evergreen plants in large-scale scale, trait-based studies" and "By generating global maps for all 16 FTs at 0.1° resolution, we have provided the most comprehensive set of trait maps based on statistical upscaling approach to date" do not seem to be enough. The different alternative explanations for some of the patterns found, while candid, are a bit vague and unsatisfactory. And one of the potentially most novel aspects of the study, the coordination of traits into successful (or not) combinations depending on climate and growth form, is just superficially, almost implicitly touched upon. This limitation might just be the result of not effective enough writing, i.e. failure to highlight explicitly enough what indeed is the major breakthrough. But it might well be that the study is great, but not groundbreaking enough for this journal. I suggest the authors are given an opportunity to address this point.

Responses:

We appreciate these encouraging comments, which we have now addressed.

In terms of the “the coordination of traits into successful (or not) combinations depending on climate and growth form”, we have included additional analyses (Table S14–S15 and Fig. S16) in our revised manuscript, and discussed the results in the Discussion (L287 – L307):

“Our analysis also revealed a differentiation in trait combinations among the three plant groups, specifically in the covariation of SSD with other FTs. SSD in non-woody plants was strongly correlated with LES traits (Fig. 1a). SSD in woody deciduous plants was relatively independent (Fig. 1b), while SSD in woody evergreen plants was more correlated with plant size traits (Fig. 1c). This novel finding is explicable in adaptive terms. In dry climates, some herbaceous and succulent plants can fix CO₂ through their green stems, reducing water loss from the leaves and providing an extra source of carbon^{61–63}. The variation of SSD was considered to be independent of other FTs in the global spectrum of plant form and function¹². Previous studies were either conducted on global vascular plants¹² or analysed trait data at the community level by using community-weighted means^{9,14}. These studies did not identify the functional differences among the non-woody, woody deciduous and woody evergreen plants, consequently failing to detect the distinctive trait combinations associated with SSD. We also extracted pixel values from the GAM-based global maps of the six major traits in each plant group and performed PCA and RDA on them to test these observed trait combinations were still present in the predicted trait values (Fig. S16). The distinct patterns of SSD among the three plant groups were not evident in the predicted trait values – particularly in non-woody plants, where the previously strong correlation between SSD and LES traits was now absent (Fig. S16a). Trait data used in original PCA and RDA were calculated from the database of Díaz et al.⁴⁰ (see Methods) in which observed records for SSD are available only for a few non-woody species; missing values of SSD were imputed via leaf dry matter content (LDMC)⁴⁰, which was found to be closely related to both LMA and N_{area}⁹: perhaps accounting for the strong correlation between (LDMC-derived) SSD and the LES traits.”

More minor points:

I have doubts about the reasons for the traits considered. Some of them are not properly justified or are at risk of being trivially correlated with each other, therefore potentially distorting the multivariate analyses. For example, leaf mass and leaf area, or leaf C (which varies very little in general and, admittedly, see L 355, among climates in particular), especially considering they also look at leaf N and leaf dry mass content. And seed length and seed mass. Whereas rationale for each (or some) of the individual traits is summarily provided, there is no justification as to why this set of traits and not any other random list of traits were considered that are directly relevant to the specific questions in this study.

Responses:

We have clarified this point here:

The six major traits are selected according to the global spectrum of plant form and function as defined in Díaz et al., 2016. Díaz and colleagues also provided species-level means of the traits in Díaz et al., 2022. We selected these traits and conducted PCA and RDA analyses to test whether the two orthogonal dimensions of covariation in plant size traits and leaf economics spectrum traits, as initially discovered by Díaz and colleagues (2016), are also applicable to each of the three plant groups we studied. We have explained that in Main (L36 – L41), Discussion (L270 – L273) and Methods (L474 – L475). The other remaining traits only have plot-level means and cannot conduct PCA and RDA within separate plant groups.

The plot-level mean values of all 16 traits are collected and calculated from the latest published sPlotOpen database. Prior studies (Bruehlheide et al., 2018 and Joswig et al., 2022 in *Nature Ecology & Evolution*) have investigated the influence of climate and soil variables on trait combinations of these traits at the community level. However, the global patterns of these traits predicted from their relationships with climate remain unexplored. Analyses of the six major traits with species-level means revealed variations in trait-trait and trait-climate relationships among the three plant groups. We thus fitted GAMs considering three bioclimatic variables and fractional cover of plant groups for all 16 traits with plot-level means to simulate the global patterns of all 16 traits. The ecological functions of our selected 16 traits were shown in Table 1. But we did not blindly adopt all the traits available in the sPlotOpen database; and we made choices based on our own considerations. For instance, we focused on N_{area} and P_{area} rather than N_{mass} and P_{mass} , as now explained in detail in the

revised manuscript (L445 – L452). We pasted the relevant statements in the responses to the next comment.

Why N and P are expressed per area and not as per mass, which is the expression used in the articles the authors use as basis, and in general more common in the trait literature? A hint is given around L. 260 but it should be made more explicit.

Responses:

We have now clarified the reasons for the selection of N_{area} and P_{area} in Methods (L445 – L452):

“Here we focus on N_{area} and P_{area} , because Osnas et al.⁷⁹ showed that leaf nitrogen and phosphorus contents are approximately distributed proportional to leaf area rather than mass. N_{area} and P_{area} were found to be proportional to LMA and photosynthetic capacity of plants^{59,60,80}. Both LMA and photosynthetic capacity are quantitatively predictable from climate^{45,80}, which may lead to potentially high predictability of N_{area} and P_{area} from climates. In contrast, values of N_{mass} and P_{mass} may be more conservative under climate change. We also conducted the following analyses on N_{mass} and P_{mass} (see Supplementary Information for details) and found that climate is not good predictor of their distributions.”

We have also provided the results of analyses on N_{mass} and P_{mass} in Supplementary Information. Please check *SI 2 Analysis for mass-based leaf nitrogen content (N_{mass}) and leaf phosphorus content (P_{mass})* for more details.

Why climatic variables explain more variance for woody than for non-woody plants (around L 270): this explanation is plausible, but the fact that woody plants have permanent organs that have to stand climate throughout the year for many years may also explain this tighter association.

Responses:

We agree and we have now added this suggested explanation into the manuscript (L312 – L316).

“One explanation is that woody plants have permanent organs that must endure climatic conditions year-round over the years. In addition, short-stature non-woody plants are closer to the ground and experience different physical conditions compared with tall-statured woody plants⁴²;

therefore, the microclimate of non-woody plants in the understorey cannot be directly represented by macroclimate variables⁴².”